# Mechanisms and scenarios of the unprecedent flooding event in South Brazil 2024

Leonardo Laipelt[1], Fernando Mainardi Fan[1], Rodrigo Cauduro Dias de Paiva[1], Matheus Sampaio[1], Walter Collischonn[1] and Anderson Ruhoff[1]

[1]Institute of Hydraulic Research (IPH), Federal University of Rio Grande do Sul, Porto Alegre, Brazil

*Correspondence to*: Leonardo Laipelt (leonardo.laipelt@ufrgs.br)

**Abstract.** In May 2024, an extraordinary precipitation event triggered record floods in southern Brazil, particularly impacting complex river-estuary-lagoon systems, and resulting in unprecedented impacts on the local population and infrastructure. As climate change projections indicate an increase in such events for the region, understanding these flooding processes is essential for better preparing cities for future events like the May 2024 flood. In this context, hydrodynamic modelling is an important tool for reproducing and analysing this past extreme event. This paper presents the first detailed hydrodynamic assessment of this unprecedented flood, the worst registered natural disaster in Brazilian history. We also performed the first validation of a detailed hydrodynamic model using new observations from the SWOT satellite. The study investigates the main mechanisms that governed the disaster and assesses scenarios for hydraulic flood control interventions currently under public debate, with a focus on the most populated areas of the Metropolitan region of Porto Alegre (RMPA) capital city. The results demonstrated that the model accurately represented the event, with average NSE, RMSE and BIAS of 0.82, 0.71 meters and -0.47 meters, respectively, across the basin's main rivers. Furthermore, the simulated flood extent showed an 83% agreement with high-resolution satellite images. Our analysis of the governing mechanisms showed that the Taquari River was mainly responsible for the peak in the RMPA, while the Jacuí River contributed most to the flood's duration. Additionally, the synchronization of the flood peaks from both rivers could have increased water levels by 0.82 meters. Evaluated hydraulic interventions demonstrated that the effectiveness of the proposed measures varied by location, with a generally limited influence on RMPA water levels (lower than 0.38 m). By accurately assessing the May 2024 flood, this study enhances the understanding of a complex river-estuary-lagoon system, quantifies the impacts of adverse scenarios, and reveals the limitations of potential hydraulic structure interventions. Finally, modelling this unprecedented event offers valuable insights for future research and global flood management policies.

## 1.  Introduction

The May 2024 flood in southern Brazil is considered the worst natural disaster in Brazilian history given its magnitude, spatial coverage and impacts (Collischonn et al., 2025). The flood affected hundreds of thousands of people, displacing entire neighbourhoods, causing numerous fatalities, and inflicting widespread damage on urban infrastructure and agricultural lands. Floods in southern Brazil, situated in the sub-tropical and temperate portions of South America, have increased significantly in recent decades, a trend that has been supported by both historical data and climate projections (Ávila et al., 2016; Bartiko et al., 2019; Brêda et al., 2023; Chagas et al., 2022). Nationally, flood generation in Brazil is driven by a variety of mechanisms. These include intense convective storms causing urban flash floods (Cavalcante et al., 2020; Lima and Barbosa, 2019; Marengo et al., 2023), persistent rainfall associated with South Atlantic Convergence Zone (SACZ) leading to large-scale riverine floods, and the influence of major teleconnections like the El Niño-Southern Oscillation (ENSO). Specifically, in the southern region, the primary drivers are often intense frontal systems that bring widespread and prolonged precipitation (Ávila et al., 2016; Damião Mendes and Cavalcanti, 2014). Moreover, climate change is intensifying this scenario by increasing hydroclimate and hydrological volatility and altering flood-generating mechanisms (Hammond et al., 2025; Stevenson et al., 2022; Swain et al., 2025). This, in turn, increases the frequency and severity of floods, particularly through compound events (Heinrich et al., 2023; Hendry et al., 2019; Leonard et al., 2014).

The May 2024 floods in southern Brazil primarily impacted the state of Rio Grande do Sul State (RS), including its capital, Porto Alegre. Observed rainfall data indicated precipitation exceeded 500 millimetres within a two-day period, while some locations accumulated up to 900 millimetres over 35 days (Collischonn et al., 2024). Therefore, flood reached record-breaking levels in numerous cities within the Patos Lagoon basin, which covers half of the state.

Several impacts on the population and urban infrastructure were concentrated in the Metropolitan Region of Porto Alegre (RMPA), home to over 4 million people (nearly 40% of the state's population). According to official government surveys, approximately 300,000 people in RMPA were directly affected by the flooding. The situation was significantly exacerbated by the failure of local flood protection systems. Given these devasting impacts, there is a critical need to employ advanced assessment tools, such as hydrodynamic modelling, to better understand and manage such extreme events.

The Patos Lagoon basin, most severely impacted by the May 2024 disaster, is a unique natural system. Its complex watershed combines fast-flowing mountainous rivers with a large, slow floodplain rivers. These tributaries converge into a relatively short, wide body named Guaíba River (which is also called lake due to its physical characteristics). The Guaíba, in turn, flows into the Patos Lagoon itself, an extensive water body known as the world's largest choked lagoon (Kjerfve, 1986).

After the disaster, significant public and technical debated emerged regarding the hydraulic drivers of the flood. Question focused on the relative influence of upstream rivers, the slopes generated by water inflows, and the restrictive nature of the lagoon's single outlet to the ocean. Specifically, public and governmental debates have hypothesized that additional artificial outlets could have mitigated flooding in upstream areas (Hunt et al., 2024; Silva et al., 2024a). From our understanding, the

tool to answer some of those questions is a hydrodynamic model capable of properly representing the system. This model must be properly validated using not only gauge observations but also state-of-the-art remote sensing data.

Hydrodynamic modelling has been widely used for flood assessment, including models such as HEC-RAS (USACE, 2016), LISFLOOD-FP (Bates and De Roo, 2000) and Delft3D (Lesser et al., 2004). These physically-based models represent water flow across natural systems, accurately predicting flood propagation and extent (Ming et al., 2020; Paiva et al., 2013;

Timbadiya et al., 2015). Their ability to reproduce floods in high-resolution makes them ideal for reconstructing past events and simulation various scenarios (Bates et al., 2003; Fewtrell et al., 2011; Marks and Bates, 2000). For instance, these models are particularly useful for studying complex interactions in medium-to-large basins (O'Loughlin et al., 2020; Paiva et al., 2013), where precipitation is expected to become more concentrated. In these coupled systems, the synchrony between the peak flows of major tributaries and the estuary–lagoon water level is a primary determinant of flood severity, directly informing

the timing and feasibility of structural and operational measures (Guse et al., 2020). While previous studies have often focused on individual rivers or local interventions (Dutta et al., 2007; Patel et al., 2017; Timbadiya et al., 2015; Zarzuelo et al., 2015), few have examined synchrony and mitigation within an integrated, river–estuary–lagoon framework at regional scale. Moreover, simulating flood mitigation scenarios is essential for evaluating interventions, defining optimal locations for new structures, assessing the efficiency of existing ones (Abdella and Mekuanent, 2021; Ghanbarpour et al., 2013; Zhang et al.,

2021), and identifying areas of high risk (Cai et al., 2019; Li et al., 2019; Masood and Takeuchi, 2012).

Among different applications, Neal et al. (2011) assessed the 2007 United Kingdom floods using the LISFLOOD-FP two-dimensional model to simulate water levels in urban areas, validating flood extent with satellite imagery. Marengo et al. (2023) utilized the HEC-RAS 2D model to examine flood inundation extent resulting from an extreme precipitation event in Recife, Northeast Brazil. In southern Brazil, there are few studies using hydrodynamic models to evaluate historical water levels in

the Patos Lagoon basin (Alves et al., 2022; Fernandes et al., 2001, 2002; Möller et al., 1996). Alves et al. (2022) evaluated the ability of large-scale models to generate flood maps, comparing results with satellite imagery and flood extent from 2D hydrodynamic flood simulations. Fernandes et al. (2001) calibrated and validated the TELEMAC-2D model to simulate water levels over the Patos Lagoon, finding good agreement with observational data.

This study develops the first detailed hydrodynamic assessment of the unprecedented flood that occurred in 2024 in south

Brazil, which represents the worst disaster in Brazilian history. In addition to this novelty, it is the first study to utilize SWOT satellite altimetry data for model evaluation. Our primary goals are to investigate the main mechanisms governing this flood disaster and to assess hydraulic intervention scenarios for flood control in the region, which are currently under public debate. To achieve this, we address urgent and unresolved questions raised by the May-2024 flood regarding: (a) the relative influence of tributary inflows on RMPA water levels and inundation, (b) the consequences of potential peak synchrony between the

main rivers, and (c) whether additional lagoon–ocean outlets or channel operations would have mitigated upstream flooding. Prior studies did not jointly address these system-scale dynamics due to limited integrated datasets and validation across the river–estuary–lagoon continuum. Leveraging detailed bathymetry, Acoustic Doppler Current Profiler (ADCP) transects, continuous gauges, satellite flood extent, and SWOT altimetry (Biancamaria et al., 2016; Fu et al., 2024), we develop and

validate a 2D hydrodynamic model to quantify mechanisms and test counterfactual scenarios. This design yields decision-relevant evidence for stakeholders and government agencies seeking to enhance protection in the most affected areas, and ultimately allows comprehension of how this unique natural system works under extreme conditions. The insights from this study are therefore highly relevant for other complex, large-scale hydrodynamic coastal and deltaic regions.

## 2. Study Area

The Patos Lagoon basin encompasses 182,000 km² (**Figure 1a**), with its headwaters situated in Rio Grande do Sul's north-central region, characterized by deep canyon valleys transitioning into vast lowlands. The primary upstream tributaries, the Taquari, Jacuí, Caí, Sinos and Gravataí rivers, converge at the Jacuí Delta, forming a substantial estuary that leads to the Guaíba River (**Figure 1b**).

The Guaíba River, an important freshwater system in RS, plays a key role in providing drinking water, supporting navigation, and facilitating irrigation for the area. Located adjacent to Porto Alegre, the state capital, it has an average depth of 2 meters, with certain spots reaching over 30 meters near its outlet to the Patos Lagoon. Spanning approximately 10 km in width and 50 km in length, the river covers roughly 480 km².

Water from the Guaíba River flows into the Patos Lagoon, which stretches across a considerable area (250 km in length and 40 km in width), with an average depth of 5 meters, before connecting to the coastal ocean. Consequently, tidal fluctuations influence the downstream water levels in the Patos Lagoon basin. Furthermore, wind forces significantly impact both the Patos Lagoon and the Guaíba River, with prevailing winds oriented in a NE-SW direction across the state. The wind changes the water level in both systems, with SW (NE) wind restricting (facilitating) Guaíba flow into the Patos Lagoon, increasing (decreasing) Guaíba water levels up to 50 cm (Collischonn et al., 2025; Laipelt et al., 2025).

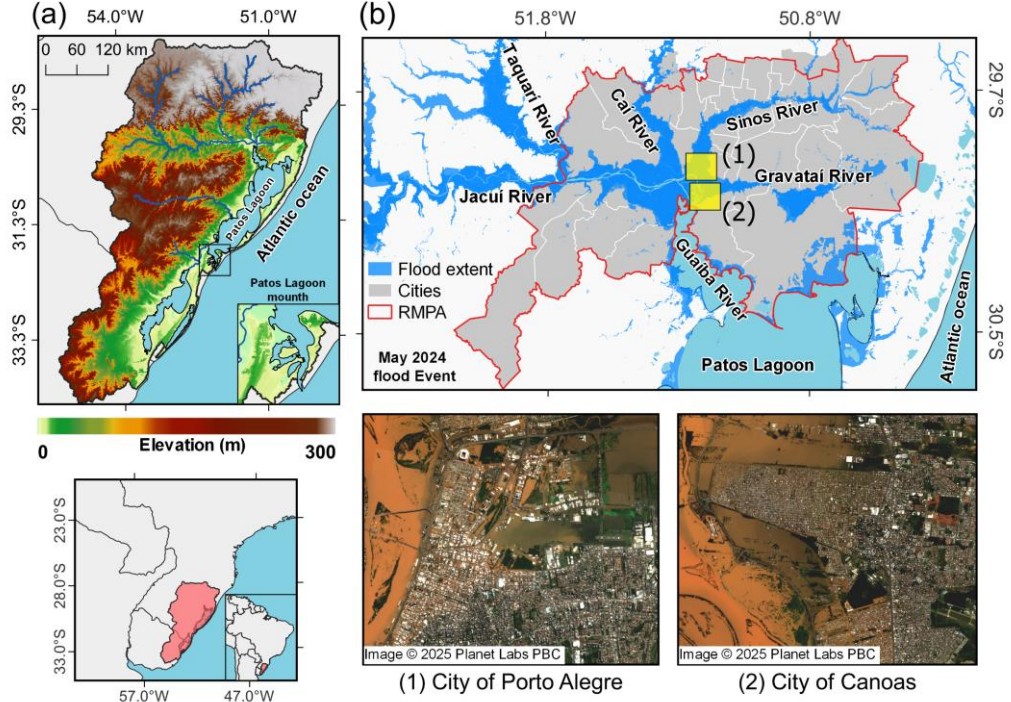

**Figure 1**: The Patos Lagoon basin, located in southern Brazil, features a mouth that connects to the Atlantic Ocean (a). The basin is fed by several primary tributaries, including the Taquari, Jacuí, Gravataí, Sinos, and Caí (b). These rivers experienced unprecedented flooding during the event in May 2024, affecting densely populated areas such as the state's capital, Porto Alegre, and cities in the Metropolitan Region of Porto Alegre (RMPA).

## 3. Material and methods

### 3.1. Workflow overview

**Figure 2** summarizes the study's methodological workflow. We simulated the May 2024 flood using the HEC-RAS software (version 6.4.1) (USACE, 2016), which was driven by publicly available data (topography, bathymetry, upstream inflow and weather).

First, a calibrated baseline model was established to accurately represent the event. Using this baseline, we then designed several experiments to better understand the basin's hydrodynamic processes during an extreme flood event. These steps are detailed below.

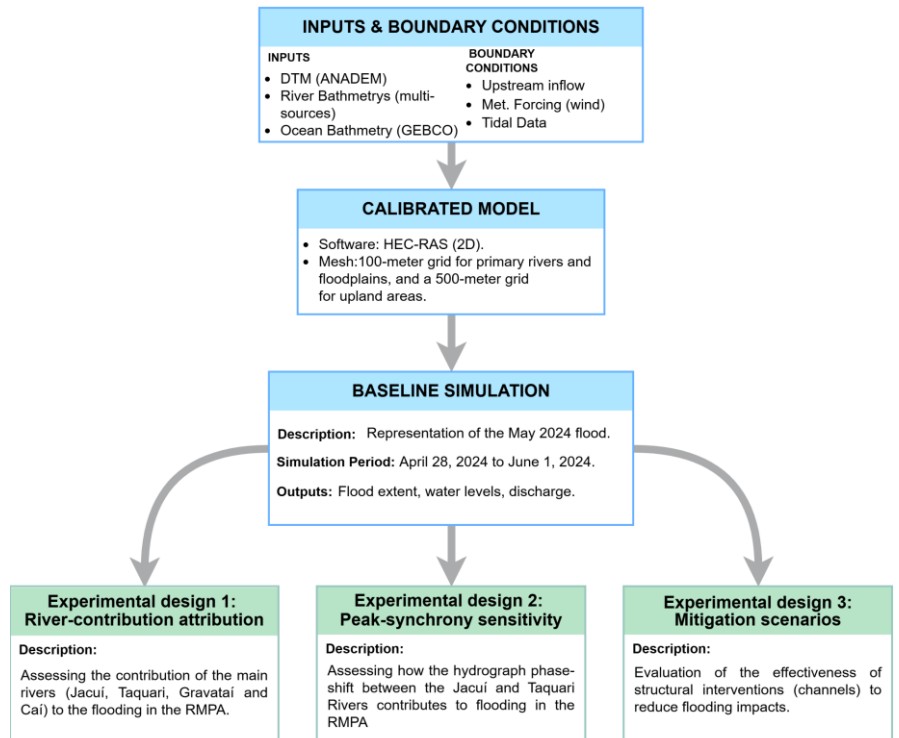

**Figure 2**. The workflow for this study proceeded by defining the input and boundary conditions for the May 2024 flood, followed by calibrating and validating a baseline model, which was then utilized to analyse multiple scenarios.


## 3.2. Simulation domain, boundary conditions and inputs

The simulation employed a two-dimensional (2D) hydrodynamic model based on the shallow water equations, accounting for inertia, gravity, friction, pressure, turbulent viscosity, wind and Coriolis effects. The model domain covers 23,000 km² (**Figure 3b**), corresponding to a 650 km reach from the upstream Jacuí to the Patos Lagoon's ocean outlet. This extent was defined based on available streamflow observations from the flood event. To ensure accuracy at the downstream boundary, the domain also includes a 45 km portion of the ocean, which is necessary to properly represent tidal effects influencing the lagoon.

Upstream boundary conditions were defined using streamflow time series for the main rivers of the Patos Lagoon basin, acquired from the Brazilian Water and Sanitation Agency (ANA) network (https://www.snirh.gov.br/hidrotelemetria/, accessed in July, 2024) (**Figure 3a**; **Supplementary Table 1**). The downstream boundary condition used tidal level time series from the Brazilian coast monitoring system (SIMCosta) network (https://simcosta.furg.br/home, accessed in August, 2024), located at the Patos Lagoon mouth. This ensures that the representation of the water levels over the basin are realistically subjected to the backwater effects of the tides under variable marine conditions. We did not incorporate auxiliary data for the minor tributaries, as their contributions were considered negligible compared to the primary river flows.

The simulation period ranged from April 28, 2024, to June 1, 2024, including a 10-day initial period for model warm-up. We

implemented a variable-resolution mesh, using a 100-meter grid for primary rivers and floodplains, and a 500-meter grid for upland areas to optimize computational efficiency.

Surface topography was represented using the ANADEM Digital Terrain Model (DTM) product from ANA. ANADEM is a freely available DTM for South America derived from the COPERNICUS GLO-30 DEM (AIRBUS, 2020), removing vegetation bias using machine learning and satellite altimetry data (Laipelt et al., 2024).

To improve the hydraulic representation, the ANADEM DTM was modified to incorporate bathymetry from multiple sources. For the Jacuí, Gravataí, Sinos and Caí rivers, we used an interpolated bathymetry based on cross sections from publicly available data, as developed and validated by François (2021). For the Taquari River, where data was insufficient, the DTM was adjusted by lowering the channel elevation by approximately 4 meters between the upstream boundary condition and its confluence with the Jacuí River. Bathymetry for the Guaíba River and Patos Lagoon was derived from digitized nautical charts

provided by the Board Hydrography and Navigation of the Brazilian Navy. Finally, ocean bathymetry from the General Bathymetric Chart of the Oceans (GEBCO) version 2024 (GEBCO, 2024) was integrated into the DTM to improve the representation of tidal dynamics in the Patos Lagoon estuary.

Wind data from the National Meteorological Institute of Brazil (INMET) (https://bdmep.inmet.gov.br/, accessed in August, 2024) were incorporated, as wind significantly impacts water levels in the Patos Lagoon and Guaíba River (Laipelt et al.,

2025). The effect of the wind was computed using a drag formulation (Hsu, 2003), which modifies the aerodynamic roughness length by assuming a logarithmic velocity profile.

Initial values of Manning's roughness coefficient were derived from the literature and refined through manual calibration for the study period. Previous studies have modelled Patos Lagoon basin using Manning coefficients between 0.015 to 0.04 (António et al., 2020; Hillman et al., 2007; Marques et al., 2009; Martins and Fernandes, 2004), while values for the Guaíba

River have ranged from 0.025 to 0.040 (Marques et al., 2009; Possa et al., 2022; Seiler et al., 2020). In the absence of specific data, the calibration of Manning's coefficient for the main channel was based on general hydrodynamic applications (Chow, 1959). For the floodplain, Manning's roughness coefficient values were calibrated by testing a range from 0.05 to 0.15, following the HEC-RAS manual guidelines (USACE, 2016).

A set of Manning's values evaluated for the 2D hydrodynamic model is presented in **Supplementary Table 2**, with statistical

performance demonstrated in **Supplementary Figure 1**. Group 1 was selected for the simulations due to its higher accuracy, with calibrated values varying from 0.025 to 0.035, for the main channels, and 0.08 to 0.15 for floodplains.

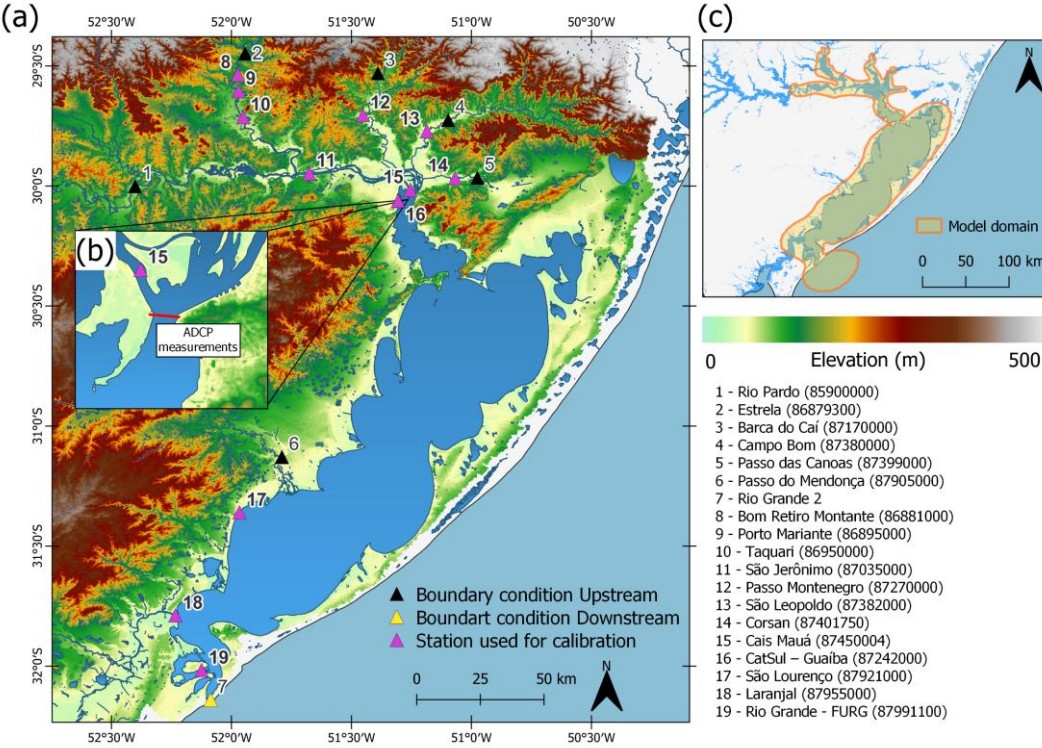

**Figure 3**: The locations and corresponding names of the gauge stations used for upstream conditions (black markers), downstream conditions (yellow markers), and calibration/validation (pink markers) are shown in (a). The ADCP measurement's location is identified by the red line in (b). The delineated geometry area (yellow) used for the two-dimensional simulation is shown in (c).

### 3.3. Observational datasets and validation metrics

To evaluate model accuracy, we validated water levels, streamflow, and flood extent. Water level validation utilized time series from independent gauge stations from ANA (**Figure 2a**; **Supplementary Table 3**), which were not used for model boundary conditions. We incorporated observations from the Surface Water and Ocean Topography (SWOT) mission (Biancamaria et al., 2016; Durand et al., 2010; Fu et al., 2024), as SWOT has been previously shown to adequately represent this event (Laipelt et al., 2025). SWOT, a CNES/NASA collaboration, is a radar interferometry sensor and provides instantaneous observations of water surface elevation (WSE) and water slope for rivers wider than 100 meters, with a revisit frequency of 21 days.

The simulated flood extent accuracy was verified against a high-resolution (5 m), clear-sky Planet RapidEye image captured near the flood peak on May 6, 2024 (Planet Team, 2024). The observed flood extent was then determined by calculating the Normalized Difference Water Index (NDWI) using **Equation 1**:

$$NDWI = \frac{(\alpha_{green} - \alpha_{infrared})}{(\alpha_{green} + \alpha_{infrared})} \tag{1}$$

where $\alpha_{green}$ is the green band and $\alpha_{infrared}$ is the infrared band.

To ensure compatibility, we resampled both the satellite flood extent and the model simulation output to a common 30-meter grid. We then calculated the relative extent ratio (**Equation 2**) between the simulate flood extent and observed by the satellite data.

$$H = 100\% \frac{P_{sim} \cap P_{obs}}{P_{obs}} \tag{2}$$

where $P_{sim}$ is the number of flood pixels obtained from the simulation and $P_{obs}$ the number of flood pixels identified with satellite imagery.

Simulated streamflow accuracy was evaluated against six field measurements obtained using an Acoustic Doppler Current Profile (ADCP) instrument between May 5, 2024, and May 31, 2024 (Andrade et al., 2024; Silva et al., 2024b) collected near the Cais Mauá station (**Figure 2c**; **Supplementary Table 4**).(Andrade et al., 2024). Measurements on May 5 and 6 were

obtained at the cross section of "Ponta da cadeia", while subsequent measurements (May 9 to 31) were at the "Ponta do Dionísio" cross-section, 8 km downstream. The "Ponta da cadeia" measurements are likely underestimates of total flow, as significant overbank flow was bypassing this cross-section on those dates.

Model performance for water level was quantified using the Root Mean Square Error (RMSE), the Nash-Sutcliffe Efficiency coefficient (NSE) and the Mean Average Error (BIAS), as showed in **Equations 3**, **4** and **5**, respectively. To compare with the

observations, the water level time series of the model were extracted from the simulation at the locations corresponding to the gauge stations.

$$RMSE = \frac{1}{2}\sqrt{\sum_{i=1}^{n}(O_i - P_i)^2} \tag{3}$$

$$NSE = 1 - \frac{\sum_{i=1}^{n}(O_i - P_i)^2}{\sum_{i=1}^{n}(O_i - \overline{O_i})^2} \tag{4}$$

$$BIAS = \frac{1}{n}\sum_{i=1}^{n}(O_i - P_i) \tag{5}$$

where $O_i$ is the observed and $P_i$ the predicted values and $n$ the number of samples.

### 3.4. Experimental design

### 3.4.1. Baseline simulations

The baseline simulation represents the calibrated model of the May 2024 flood event, incorporating all observed data, including

river inflows, tides, wind, and bathymetries described in Section 3.1.

### 3.4.2. River-contribution attribution

We used attribution scenarios to evaluate how individual tributaries influence flooding in the RMPA region. Specifically, we simulated four scenarios based on the baseline model, each excluding the streamflow from one main river (Jacuí, Taquari, Sinos, or Caí) to quantify its specific impact on water levels. This approach was selected because flood wave propagation is a non-linear process in the system, with its velocity governed by the system's channel geometry, roughness, and by the influence of the floodplains.

### 3.4.3. Peak-synchrony sensitivity

This analysis evaluated the combined flood impact of the Jacuí and Taquari rivers on the RMPA. Although these tributaries have distinct flow propagation times, their peaks can synchronize depending on the spatio-temporal distribution of rainfall. In May 2024, the region was impacted by a sequence of two cold fronts between April 27 and May 2. Such sequential atmospheric events can lead to peak synchronization if the first system triggers discharge in the slower-responding basin (Jacuí), while a subsequent system impacts the faster basin (Taquari) with a delay that matches the difference in their routing times.

To evaluate the potential consequences of such a meteorological alignment for the region's flood protection systems, we simulated a theoretical worst-case scenario by manually advancing the upstream hydrograph, used as the boundary condition for Jacuí River (at Rio Pardo), by approximately 4 days to force their flood peaks arrive simultaneously. This synchronization allowed us to evaluate the potential consequences for the region's flood protection systems.

### 3.4.4. Mitigation scenarios

We tested proposed mitigation interventions currently under public and environmental agencies debate. Specifically, these proposals, which have not yet been formally evaluated, suggest the construction of new channels to reduce regional water levels in RMPA (DRRS, 2024; Hunt et al., 2024). Although these projects are still in the conceptual stage, we used the 2D hydrodynamic model to test their potential effects. This assessment aims to better comprehend the dominant forces controlling the system's dynamics.

For each proposed intervention, we evaluated three channel widths: 100-meter, 250-meter, and 500-meter. The channels were assigned a Manning's roughness of 0.02, which corresponds to standard values for earthen channels, and were integrated into the DTM. Additionally, a refined 50-meter mesh was used in these intervention areas.

The following scenarios for flood control were tested:

**Channel connecting Jacuí to Guaíba**: This experiment (**Figure 4a**) investigated the construction of a 7000-meter-long open channel connecting the Jacuí and Guaíba rivers. Its depth was adjusted based on the upstream and downstream river bathymetric data.

**Channel connecting Patos Lagoon to the Ocean**: This experiment (**Figure 4b**) investigated the efficiency of a hypothetical 17,000-meter-long and 10-meter-deep open channel connecting the northeastern part of the lagoon to the Atlantic Ocean. The

downstream boundary for this channel used a tidal time series from the nearby SIMCosta network (tide gauge: Tramandaí; lon: -50.128; lat: -30.005) unaffected by the flood. Because this gauge lacks a precise altimetric reference, we estimated a correction factor by comparing its data to the referenced Rio Grande gauge (at the main lagoon outlet) during a non-flood period (May 2023–March 2024).

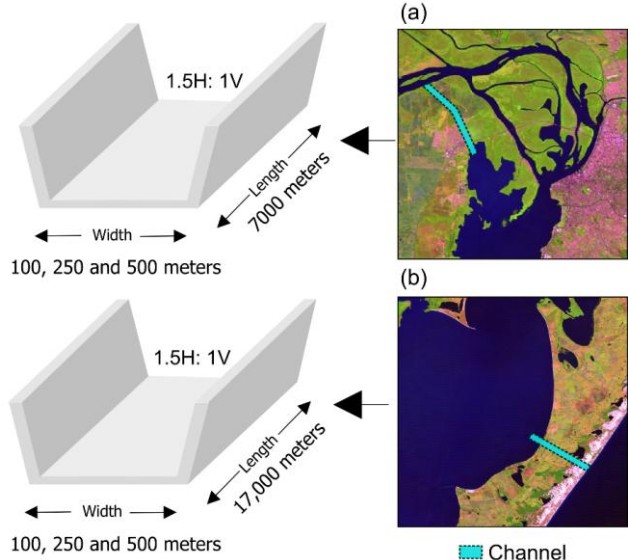

**Figure 4**: Evaluation of two hydraulic structures scenarios for flood control in RMPA: (a) an open channel connecting Jacuí and Guaíba rivers, and (b) a channel connecting Patos Lagoon to the Atlantic Ocean.

## 4. Results

### 4.1. Model Validation

#### 4.1.1. Water level

**Figure 5** shows a comparison between simulated water levels from the two-dimensional hydrodynamic model and measurements from gauge stations and SWOT. The simulation accurately captured the flow peak at most stations, showing an average BIAS of -0.47 meters in the water level peak. The bias ranged from -2.17 meters in the Bom Retiro station to 0.36 meters in the Taquari station. Conversely, the simulation failed to capture low flow at the Taquari River stations, exhibiting difference of -3.5 meters and -3.4 meters for Bom Retiro and Porto Mariante stations, respectively. This suggests an inconsistency in the representation of the river's main channel in the DTM.

Performance metrics revealed an average NSE of 0.82, with a range of 0.68 to 0.92. The RMSE averaged 0.71 meters, with values ranging from 0.10 to 1.68 meters. Differences between simulated and observed water levels exhibited relatively low BIAS, ranging from -0.2 to 0.82 meters, with an overall average below 0.07 meters. An exception was noted at the Cais Mauá gauge station, where official records diverged from the simulation after May 15. This discrepancy is explained as a

measurement error resulting from the emergency relocation of the station due to flood damage (Collischonn et al., 2024). To ensure accurate flood representation at the location, the simulation was also compared with data from a nearby experimental station (https://www.tidesatglobal.com/, accessed in September 2024). This comparison yielded a consistent agreement with the overall results (NSE = 0.85; RMSE = 0.32 meters; BIAS = -0.12 meters). Nevertheless, the metrics obtained demonstrated the essential requirements for a hydrodynamic model capable of providing locally relevant estimates (Fleischmann et al., 2019).

Additionally, the study compared simulated water levels with SWOT observations during the flood event. Both representations aligned well with water level observations from stations. The comparison between the simulation and SWOT observations revealed an average BIAS of 0.13 meters. SWOT data effectively captured water level variations along the main rivers of the Patos Lagoon basin, providing valuable validation for hydrodynamic simulations. The Corsan station proved to be an exception, as SWOT observations did not effectively capture water level during the event. **Figure 6** displays a profile line obtained by SWOT on May 6th, (near the peak in the RMPA), compared with the simulation for the same overpass (~11h a.m.) between the Jacuí River and Guaíba rivers. Both results exhibited similar water slopes, with SWOT observed a slope of 7.05 cm/km, while the 2D simulation indicated a slope of 6.45 cm/km. These findings underscore the severity of the flood event, as the water slope increased by almost 10 times compared to low-flow conditions.

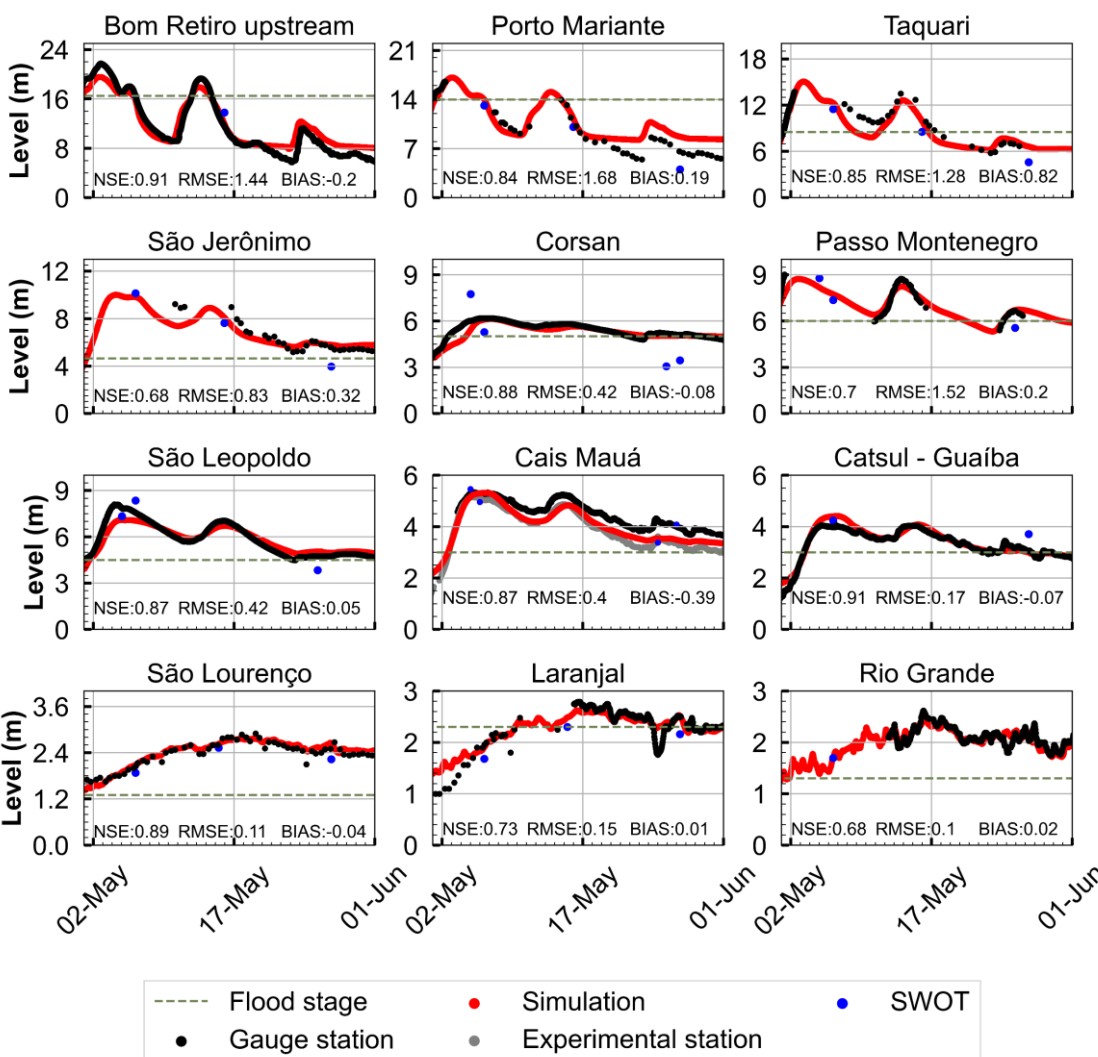

**Figure 5:** Water level simulations (red) compared to observations at twelve-gauge stations observations (black) located in the study area, alongside water surface elevation (WSE) data from the SWOT mission (blue). The grey line in the Cais Mauá plot corresponding to the experimental site data, and the dashed line indicates the flood stage at each station.

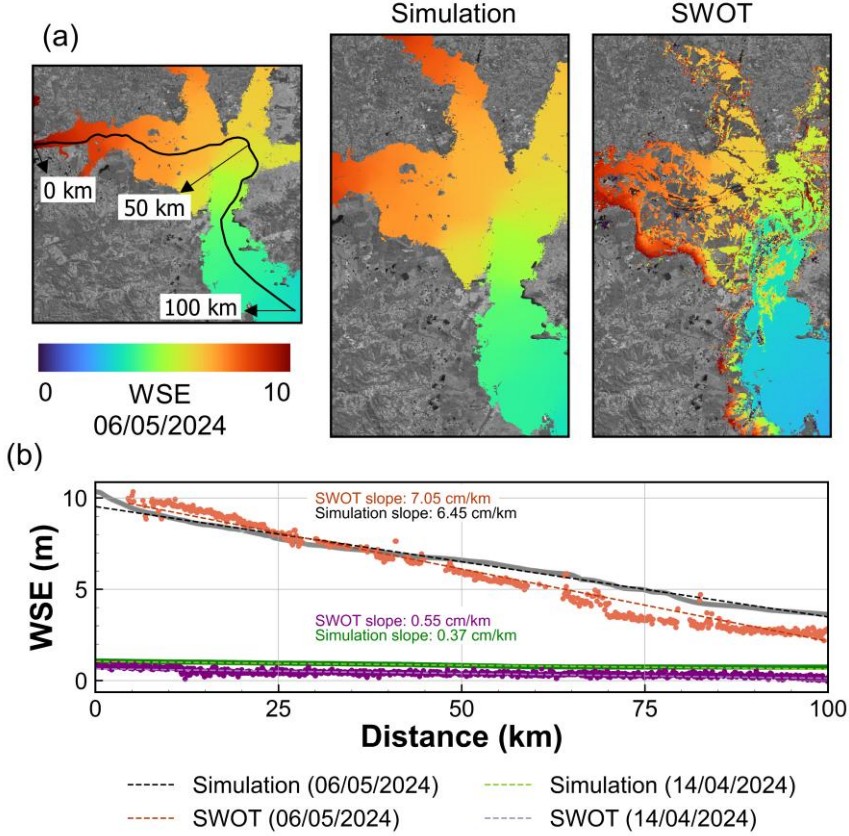

**Figure 6**: Water surface elevation slope from the 2D simulation (a) compared with SWOT data observation from May 6th. The profile line between Jacuí and Guaíba River (b) showed steep water slope changes (SWOT = 7.05 cm/km and Simulation = 6.45 cm/km) compared to low waters stable conditions (SWOT = 0.55 cm/km and Simulation = 0.37 cm/km).

### 4.1.2. Flood extent

The comparison between the flood extent captured by high-resolution optical satellite imagery and that generated by the 2D simulation is presented in **Figure 7**. While our validation was constrained to the flood extent within the simulation boundaries, excluding peripheral areas affected, the model exhibited strong agreement with the flood extent observed in the May 6 satellite images, achieving 83% agreement. However, this validation approach has certain limitations due to variations in peak flow across the basin. For instance, while the satellite image timing aligns closely with the peak water levels in the RMPA, it was captured approximately 5 days after the upstream peak (e.g., Taquari River).

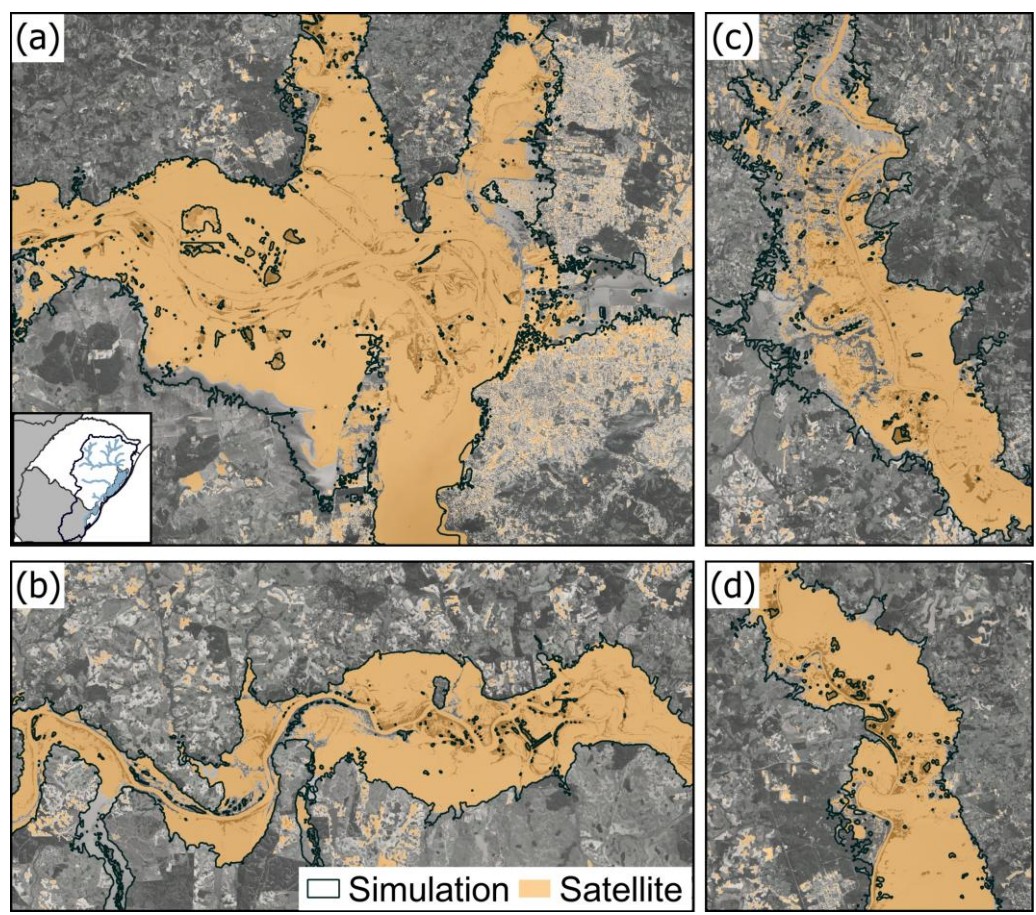

**Figure 7**: Validation of the flood extent using optical satellite images captured by the PlanetScope constellation on May 6, which was near the peak water levels in the RMPA. The simulated flood extent (indicated by the black line) closely matched the satellite observations. Illustration detail areas in the RMPA (a), the Jacuí River (b), the Taquari River (c) and the Caí River (d).

### 4.1.3. Streamflow

Simulated streamflow was compared to field measurements collected during the flood event, as shown in **Figure 8**. Measurements were collected from two nearby locations "Ponta do Dionísio" and "Ponta da cadeia" (**Figure 8a**). Due to the proximity of both sections (approximately 6 km), and the similarity of their measurements, we only compared simulation results obtained at the "Ponta do Dionísio" location. The results indicated that both the magnitude and temporal progression of the simulated streamflow were similar to observations (**Figure 8b**). The observed peak streamflow was 30,180 m³/s on May 5 at 6:00 PM, while the simulation estimated 30,724 m³/s for the same time. The model's overall bias error was 1088 m³/s, corresponding to a percentage error of 5.4%. This error is below the expected uncertainty for measurements under such extreme

conditions (McMillan et al., 2018). For individual sections, the average errors were 1062 m³/s and 1115 m³/s for "Ponta da cadeia" and "Ponta do Dionísio", respectively.

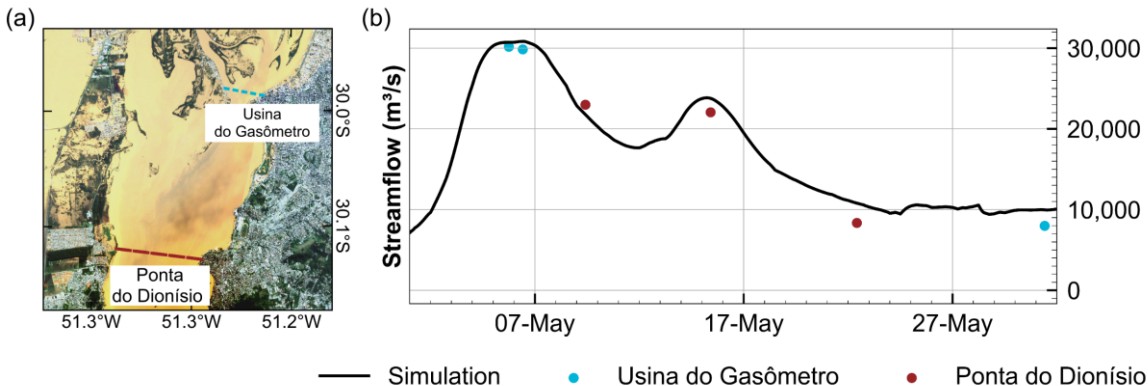

**Figure 8**: Streamflow observations collected during the 2024 flood in the Guaíba River with Acoustic Doppler Current Profiler (ADCP). Subplot (a) shows the measurements sections at Ponta do Dionísio (red dash line) and "Ponta da cadeia" (blue dash line). Subplot (b) compare these observations with the streamflow simulation data from the two-dimensional model, demonstrating good agreement.

## 4.2. Hydraulic mechanisms of the flood

### 4.2.1. River flood contribution

The analysis revealed that the Jacuí and Taquari rivers are the primary contributors to RMPA flooding, while the remaining rivers that flow into the Guaíba River (Caí, Gravataí, and Sinos) have a minimal effect on RMPA water levels.

Our findings indicated that if the Taquari River's flood contribution were eliminated (**Figure 9**, yellow line), the maximum water level would drop from 5.30 to 4.25 meters, highlighting the critical role of the Jacuí River in RMPA flooding. This scenario would also delay the peak water level by approximately 4 days compared to the May flood event.

Conversely, removing the Jacuí River's contribution (**Figure 9**, blue line) would make the Taquari River the primary contribution. The Taquari River's flow characteristics would cause the RMPA peak to occur earlier by about one day. Under these conditions, the peak water level would be lower, dropping from 5.30 to 4.75 meters. Furthermore, the flood behaviour would exhibit two peaks, similar to the actual event (**Figure 9**, black line), which is not observed when the Taquari River's contribution was removed.

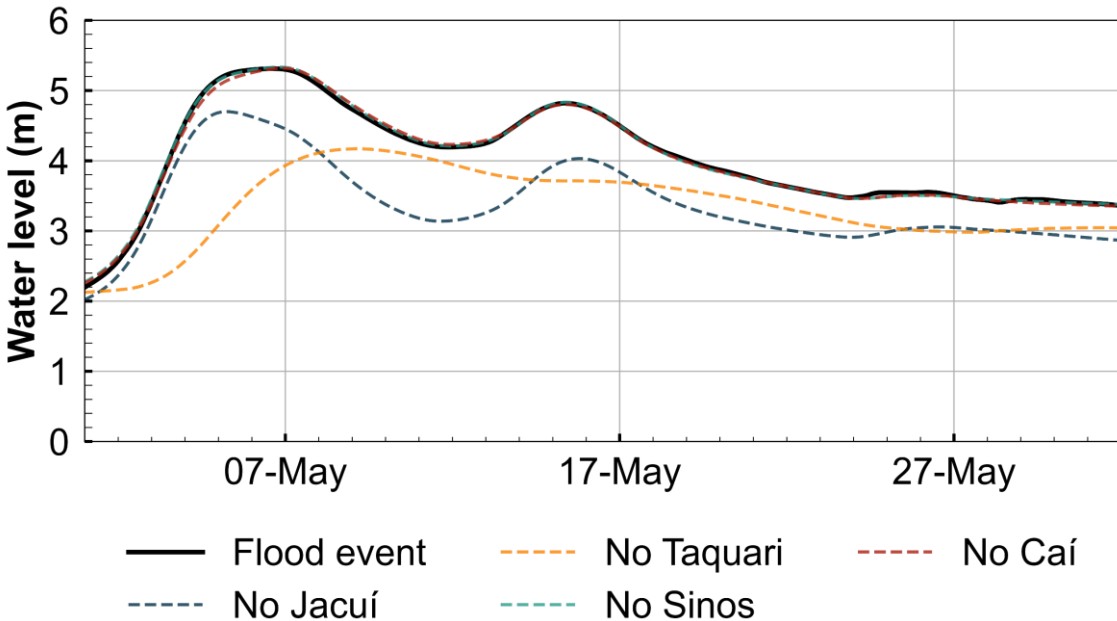

**Figure 9**: Impact of individual river contributions on RMPA flooding, based on scenarios where each river's input was removed from the simulation (Taquari, Jacuí, Sinos, Gravataí, and Caí). The Jacuí and Taquari rivers are identified as the primary flood contributions. Excluding the Taquari River's input would lower the peak flood level to 4.25 meters, while removing the Jacuí River's contribution would lower the peak to 4.75 meters. In both scenarios, the peak flood level would still exceed the flood threshold for RMPA cities.

### 4.2.2. River flood synchronization

**Figure 10** presents the simulation results for the synchronized peak flood scenario of the Taquari and Jacuí rivers. This scenario indicates an increase in the peak water level by 0.82 meters, while the flood extent would increase by 8% over the study area. In this hypothetical scenario, maximum water levels could have reached nearly 6 meters. This maximum level is equivalent to the estimated design maximum flood level of the protection system for the city of Porto Alegre.

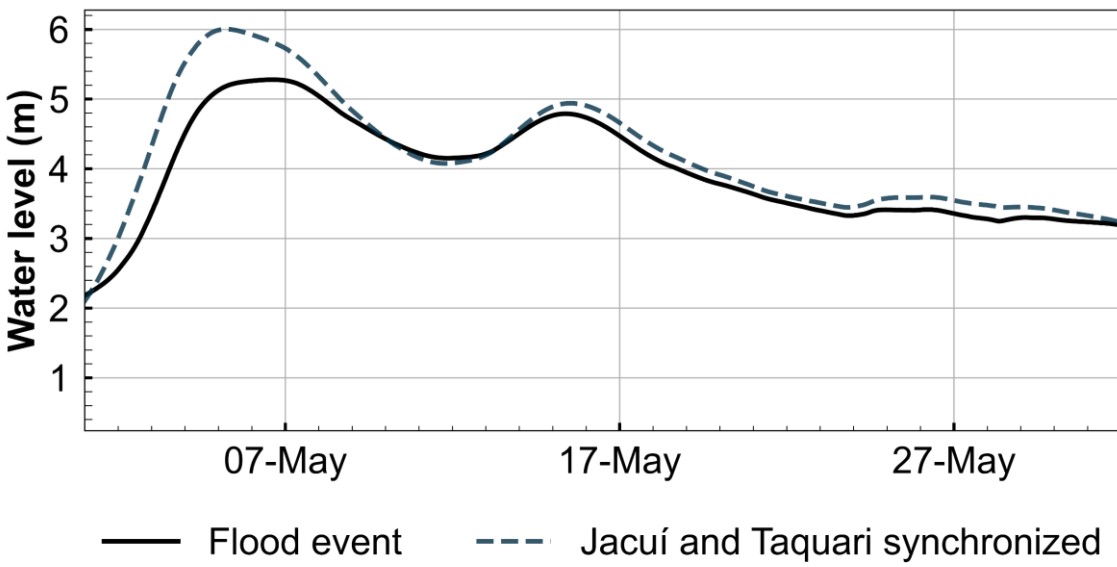

**Figure 10**. Comparative of the flood event water level simulation (black line) and synchronized water level peak of the Jacuí and Taquari Rivers arriving in Porto Alegre (grey dashed line). The maximum water level peak in the RMPA increased to 6 meters, suggesting a worsened scenario that could potentially surpass flood control systems.

### 4.3. Hydraulic interventions for flood control

#### 4.3.1.    Jacuí-Guaíba channel

Results from this experiment channel connecting the Jacuí and Guaíba rivers are presented in **Figure 11**. At Point A (**Figure 11a**), located upstream of the channel structure, the maximum water level decreased by 0.10 meters for the 100-meter width scenario and 0.27 meters for the 500-meter width scenario. At Point B, in the northeastern portion of the Jacuí Delta (**Figure 11b**), the channel's effectiveness was less pronounced, resulting in water level reductions of 0.05 meters (100-meter width), 0.12 meters (250-meter width), and 0.19 meters (500-meter width) across the evaluated scenarios. At the Cais Mauá station (Point C), representing the Porto Alegre reach (**Figure 11c**), the peak flow showed negligible variation, with the maximum water level decreasing by less than 0.10 meters in all tested scenarios.

Overall, the proposed structural intervention to lower water levels over the Guaíba River and the Jacuí Delta seems ineffective. The simulated reductions promoted by this intervention are small compared to the water level variation during the flood and compared to the model´s uncertainty range. Even the widest channel configuration—which require the most significant engineering intervention and financial investment— would result in only minor reduction in water levels, with minimal impact on flood-prone areas.

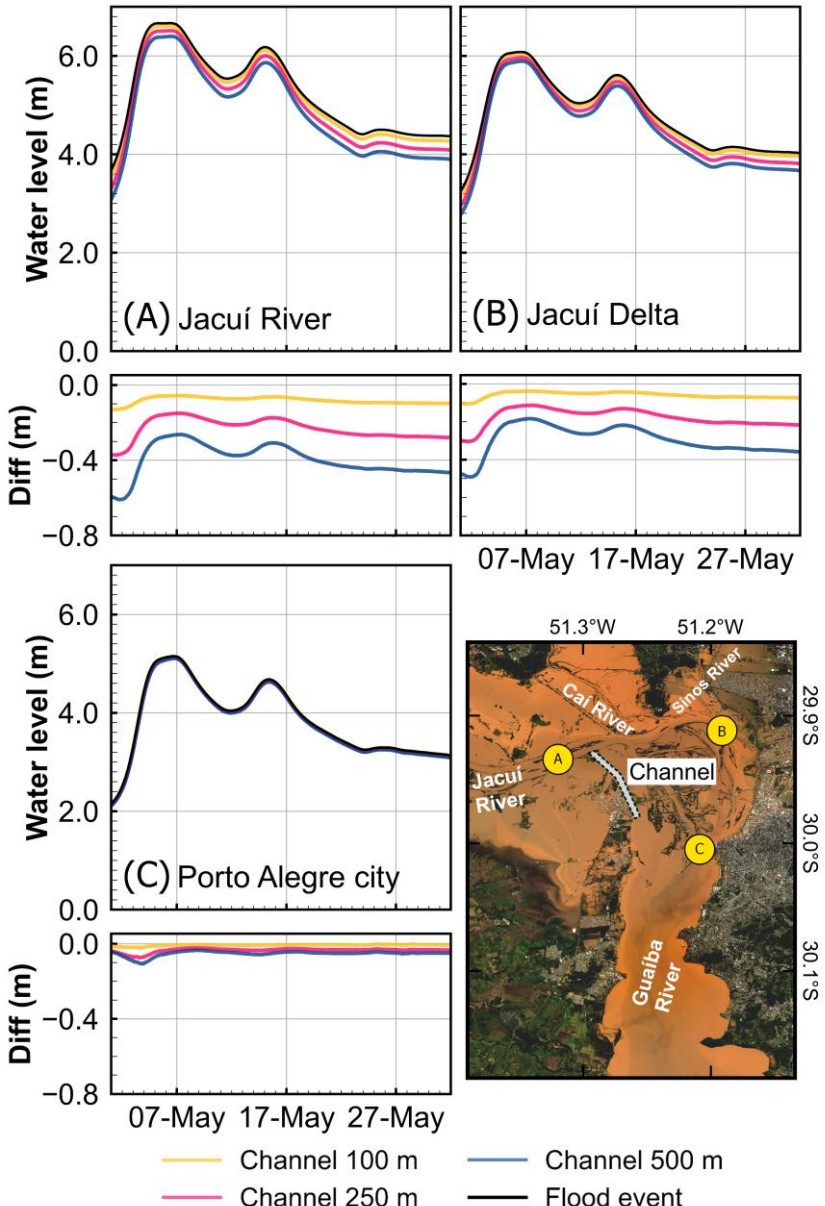

**Figure 11**. Evaluated scenario for mitigating flooding in the RMPA involved a channel connecting the Jacuí and Guaíba rivers.
Points A, B, and C illustrate the channel's effectiveness in reducing water levels around the Jacuí Delta.

### 4.3.2. Patos Lagoon channel

According to the model results (**Figure 12**), the construction of a new channel connecting the Patos Lagoon to the Atlantic Ocean could lower the average water levels within the lagoon, while having minimal impact on the maximum flow peak and

the extent of flooding upstream in the Guaíba River. At the Point C (Cais Mauá station location), the maximum water level showed negligible change, with reductions of less than 0.10 meters across all the simulated scenarios. Conversely, the results indicate a reduction in flood duration of approximately two days compared to the May flood event (**Figure 12a**).

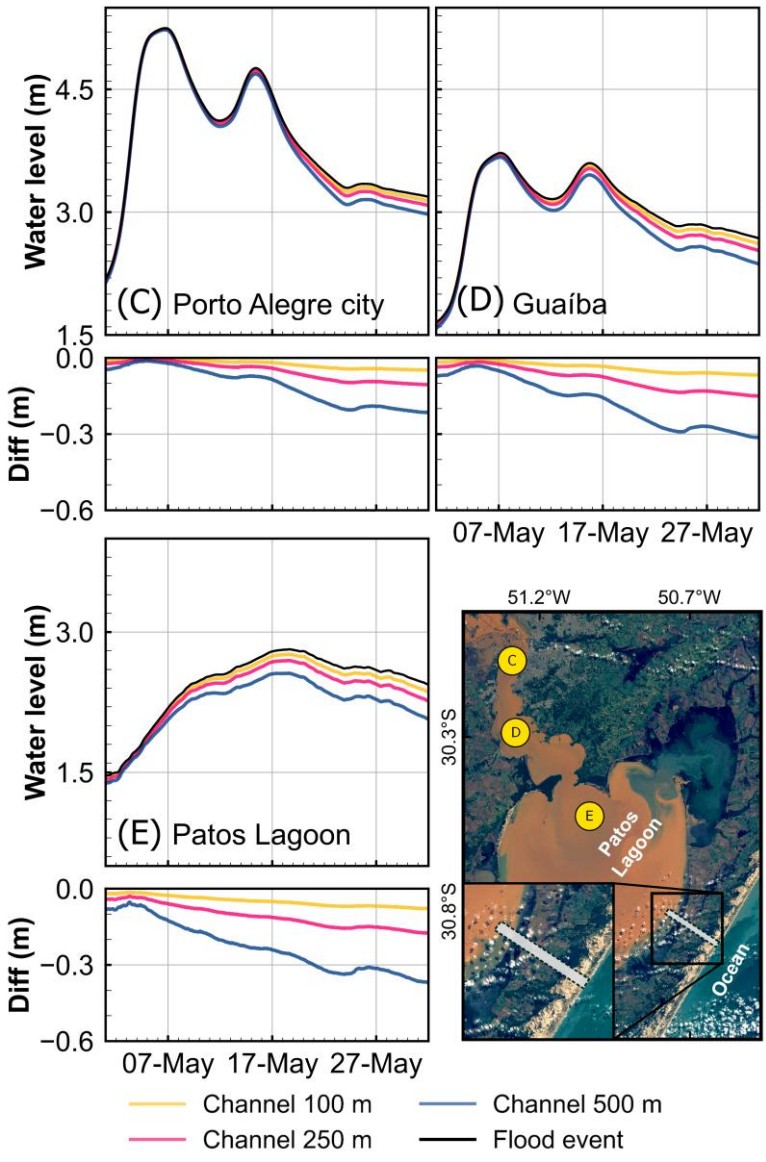

**Figure 12**. Second evaluated flood mitigation scenario for the RMPA considered the construction of a channel connecting the Patos Lagoon and the Atlantic Ocean in tits northeastern portion. Points C, D and E illustrate the influence of the channel on reducing flooding in the region.

Similar results were observed at Point D (**Figure 12b**) and Point E (**Figure 12c**), where the proposed hydraulic intervention proved ineffective in reducing the water level peak, although it did contribute to a slight decrease in average water levels and flood duration. At Point D, located in the central-southern reach of the Guaíba River, the reduction in the maximum peak was minimal (comparable to that observed at Point E). However, the simulations indicated maximum decreases of 0.07 meters for the 100-meter scenario, 0.15 meters for the 250-meter scenario, and 0.33 meters for the 500-meter scenario. At Point E, within the lagoon, the maximum reductions ranged from 0.08 meters (100-meter scenario) to 0.39 meters (500-meter scenario).

Given that none of the three evaluated scenarios produced significant changes in water levels across the study area, the proposed structural measure appears ineffective for mitigating future flood events in the RMPA.

## 5. Discussion

### 5.1. Model performance compared previous studies

In this study, we calibrated and validated a two-dimensional hydrodynamic model using HEC-RAS software to simulate the extreme May 2024 flood event within the Patos Lagoon basin in southern Brazil. The model showed an overall robust performance, achieving an average RMSE of 0.71 meters, a NSE of 0.82 and a flood area reproduction rate of 83% on the validation dataset.

Regarding accuracy, the model exhibited an error of approximately -9% compared to the maximum water peak observed in the Guaíba River, and -23% over long-term water levels. This level of accuracy is relatively consistent with current studies employing 2D hydrodynamic models in moderate to large basins. Furthermore, the difference between minimum flood level and historic flood peaks in the basin is significantly higher than the average bias found. For instance, the historic floods of May 2024 (5.2 meters) and May 1941 (4.75 meters) are substantially higher than the minimum flood level of 3 meters (at Cais Mauá gauge station).

Our performance metrics are competitive with both regional and international studies utilizing similar modelling frameworks (Bhargav et al., 2025; Gomes Calixto et al., 2020; Shustikova et al., 2019). For instance, Shustikova et al (2019) reported an RMSE of 0.84 meters and a 77% flood-extent accuracy using a 100-meter subgrid mesh in HEC-RAS 2D to simulate a flood event in Italy. Similarly, Gomes et al (2021) employed HEC-RAS 2D in a northeastern Brazil and reported an 82% reproduction of the observed flood extent, which is comparable to the overall accuracy achieved in our study. Bhargav et al (2025) applied HEC-RAS to simulate floods in the lower portion of the Narmada Basin (<10,000 km²) in India, achieving an average NSE of 0.75 and an RMSE of 0.17 meters. Although their RMSE average is lower, it is important to note that our model represents an extreme flood event across a much larger system (~180,000 km²) influenced by multiple rivers inflows, highlighting the inherent challenges of hydrodynamic modelling in large basins and complex systems. Even given these conditions, we achieved RMSE values comparable to those for some of the rivers (see **Figure 5**).

Regarding studies focused on mitigation scenarios, the baseline accuracy of the proposed model is also consistent with those typically used for planning and policy evaluation (Gomes Calixto et al., 2020; Timbadiya et al., 2015). For instance, Timbadiya

(2022) calibrated a 2D hydrodynamic model to evaluate the impact of proposed structures on flooding in the Narmada River, and obtained an RMSE of 0.84 meters and an NSE of 0.79. In the same context, Gomes Calixto et al. (2020) obtained an average NSE of 0.87 performance using a 2D model to assess flood-mitigation strategies in São Paulo.

## 5.2. Uncertainties regarding the two-dimensional model

There are some limitations regarding the model design for evaluating flooding in the study region. First, the results are depended on the DTM model data and bathymetry accuracy, which may reproduce flow propagation uncertainties due to DTM errors or limitations in its spatial resolution (30 meters). We used the ANADEM model, which demonstrates some accuracy improvements compared to other publicly available DEMs such as SRTM, COPDEM and FABDEM (Laipelt et al., 2024) for South America, with simulation results showing good agreement with observation data. Second, the upstream boundary conditions used in the model's design do not cover all the tributaries rivers of the study region due to lack of in situ observations, which may lead to local underestimations. We expect that the complete set up of independent sources validations adopted, including measured streamflow's, level gauges, satellite flooded areas and water slopes serves as quality control for models' system good representation and uncertainties understanding.

On the other hand, it is important to note that there are also uncertainties regarding the observation data used for validation. A report by Paiva et al. (2025) indicated uncertainties in the extrapolation of the rating curve, particularly in locations such as Rio Pardo on the Jacuí River and at points along the Taquari River, which are localities that contributed significantly to the flood formation. These uncertainties in the stage-discharge relationship for extreme flows may explain some of the localized differences observed between the simulated and observed water levels.

Among the flood mitigation scenarios, we additionally evaluated the sensitivity of the simulation for different parameters and proposed structures. Manning's roughness coefficient for the water bodies may influence the potential of the channels to mitigate flooding in the study region. Our sensitivity analysis showed that varying Manning's roughness coefficient in the Guaíba River between 0.025 to 0.045 would influence water flow peak in the Cais Mauá station (Point C) in -0.91 meters and 0.32 meters, respectively.

Furthermore, we utilized SWOT altimetry observations to verify the model's water level accuracy and to confirm water slope during the event, yielding results that aligned well with simulations. The application of SWOT mission data proved to be a valuable resource for acquiring information in areas with limited monitoring, enhancing our comprehension of flood dynamics and serving as supplementary data for validating hydrodynamic models.

## 5.3. Recommendations for flooding management strategies in the region

Measures to mitigate flooding impacts are urgent need for locations that are experiencing an increase in in extreme flood events due to climate change (Alfieri et al., 2016; Wang et al., 2022; Wasko et al., 2021), as is the case of southern Brazil. We assessed

different flooding scenarios based on the unprecedented flood that devasted the RMPA in May 2024, focusing on its potential impact on densely populated areas.

The evaluation of river contributions and synchronized scenarios highlights the potential of hydrodynamic models for flood management, which has been demonstrated in many other studies(de Arruda Gomes et al., 2021; Bhargav et al., 2025; Gomes Calixto et al., 2020; Guse et al., 2020; Wulandari et al., 2025). Wulandari et al (2025) used HEC-RAS 2D to reproduce an

extreme flood event in Indonesia and identified their Tallo River's main individual role int the event. Similarly, Guse et al (2020) analysed large German and Austrian river systems and evaluated the consequences of synchronized tributary rivers with the main river to floods.

Our results indicate that forecasting systems should prioritize accurate predictions of these Jacuí and Taquari rivers, including their timing relative to one another, as small changes in peak alignment can produce large impacts on the RMPA, in addition

to the impact on cities alongside both rivers

Moreover, the results regarding the hydraulic interventions showed that even the most effective configuration could only reduce peak flows by nearly 13% in the capital Porto Alegre, which is insufficient to prevent flooding in the affected areas. These findings suggest that the hydraulic interventions tested in this study would be of limited benefit in reducing the flooding due to the extreme rainfall in May 2024.

In this context, studies indicates that structural measures such as dike walls, levees may not be the most effective strategies for flood mitigation in the context of climate change (Alfieri et al., 2016; Burrell et al., 2007; Serra-Llobet et al., 2022), and in some scenarios could potentially increase flood hazard (Blöschl, 2022; Ommer et al., 2024). The implementation of protective structures also encourages development and investment in high-risk areas, potentially leading to more severe consequences when its failure, phenomenon kwon as the "levee effect" (Di Baldassarre et al., 2018). For instance, Porto Alegre's flood

protection system, developed in the 1970s following historic floods in 1941 and 1967, the false sense of security encouraged increased urban development near these systems (Miranda, 2016), which were the areas most affected during the May 2024 flood (Collischonn et al., 2025). The city Canoas, the second most populated city in the RMPA, experienced similar issues, with urbanization near its dike systems resulting in failures and a high number of houses being impacted by flooding (Collischonn et al., 2025).

To reduce the consequences of extreme flood events in the Patos Lagoon basin more non-structural interventions seems relevant. These included adopting zoning policies to limit development in flood-prone areas (Poussin et al., 2012; De Risi et al., 2015; Serra-Llobet et al., 2022). For example, spatial zoning measures in the Netherlands were found to have a risk reduction capacity of 25 to 45% (Poussin et al., 2012). Other important non-structural approaches include early warning systems, flood forecasting, and efforts to increase public awareness and improve behaviour responses to floods (Alfieri et al.,

2012; Henriksen et al., 2018; Perera et al., 2020). Additionally, collaborative framework with public participation can also lead to more cost-efficient solutions to increase flood risk assessments across communities (Henriksen et al., 2018). The advantaged of non-structural measures included lower costs, greater sustainability, and easier of implementation (Dawson et al., 2011; Kundzewicz, 2002). Therefore, cities can mitigate flood risks by reducing population exposure to extreme floods

without relying solely on structural solutions (Hall et al., 2006; Majidi et al., 2019), due to their complexity and high maintained costs, and often only minimalize impacts, and are challenging to adapt to climate change scenarios (Burrell et al., 2007; Serra-Llobet et al., 2022).

## 6. Conclusion

Our study evaluated different flood scenarios in southern Brazil's RMPA region, based on historical May 2024 flood. The analysis was conducted using 2D hydrodynamic modelling, which was validated using water level, flood extent and streamflow data, demonstrating accurate representation of the May 2024 event.

Our findings are summarized as follow:

(i) The Taquari River was responsible for most of the flow peaks in the RMPA, while the Jacuí River contributed to the flood's duration. Others (Caí, Sinos and Gravataí) flowing into the Guaíba River did not significantly impact overall water levels in the RMPA, although contributed to localized flooding.

(ii) Synchronized flow peaks of the Jacuí and Taquari Rivers in the Guaíba River would have increased water levels by 0.82 meters, exacerbating the flood scenario in the RMPA. At the Cais Mauá station, water levels would have exceeded 6 meters, surpassing the threshold for the flood protection system developed for the city of Porto Alegre. This scenario, constructed using May 2024 flood conditions but advancing the Jacuí River's flow peak by approximately 4 days, presents a significant risk to the state capital and remains plausible under heavy rainfall conditions common in the region.

(iii) The proposed hydraulic structures of additional channels alternatives would not have been sufficient to prevent RMPA flooding entirely. Our results also indicated that the degree of flood mitigation structures would not have been uniform across the RMPA. This spatial disparity in performance suggests that the limited overall impact may be linked to a combination of factors, including the specific design of the interventions, local hydrogeomorphic features, and the unprecedented magnitude of the flood event itself.

The analyses reported in this study can aid decision-makers in improving flood management strategies for RMPA region, emphasizing the vital role of hydrodynamic models in predicting and evaluating hydraulic interventions, as well as identifying opportunities for non-structural measures. Future research should benefit of high-resolution (1 to 5 meters) data based on Light Detection and Ranging (LiDAR) sensors to assess the impact of urbanization on regional flooding and identify risk zones in the context of increased flooding due to climate change.

Finally, this research advances a methodological framework predicated on multi-source data integration for the robust performance assessment of hydrodynamic simulations. By incorporating multiple, independent observational datasets, we significantly enhanced the model's predictive accuracy and its fidelity in reproducing this flood event. We expect that the presented methods will serve as a reference for studies in other locations, as well as for analyses of the efficiency of structural measures for flood control.

*Data availability*

The water level data used in this study are provided by the Brazilian Water Agency and Sanitation (ANA) (https://www.snirh.gov.br/hidrotelemetria/, accessed in July, 2024). The tidal data is available at the SIMCosta platform (https://simcosta.furg.br/home, accessed in August, 2024). The meteorological data are available at (https://bdmep.inmet.gov.br/, accessed in August, 2024) by the National Meteorological Institute of Brazil (INMET). SWOT data can be accessed through the NASA Earth Data repository (https://search.earthdata.nasa.gov/search, accessed in June, 2024). The digital terrain model (DTM) is publicly available on Google Earth Engine platform.

*Author contributions*

LL, FMF and RCDP contributed to the study's conception and design. LL and FMF wrote the manuscript draft. LL performed model simulations analysed the data. RCDP, MS, WC and AR reviewed and edited the manuscript.

*Competing interests*

The author declares that there is no conflict of interest.

*Acknowledgement*

The authors would like to gratefully acknowledge the support of the Google Earth Engine team. We thank the Brazilian National Water and Sanitation Agency (ANA) and the Geological Service of Brazil (SGB) for their efforts in providing high-quality in-situ observations during the 2024 floods. We also thank CNES and NASA for making SWOT mission data publicly available. We also thank the reviewers for their valuable contribution for the final version of the manuscript.

*Financial support*

This research has been supported by the Brazilian National Council for Scientific and Technological Development (CNPq).

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
