# Peer review of "Mechanisms and scenarios of the unprecedent flooding event in South Brazil 2024"

_EGUsphere, 2025_

## Referee Comment (RC2)

**"Mechanisms and scenarios of the unprecedent flooding event in South Brazil 2024"**

The authors examine the May 2024 flood in the Patos Lagoon basin in southern Brazil by implementing HEC-RAS 2D. The hydrodynamic model is calibrated and evaluated by comparing their simulations with water level records from gauge stations along the tributary rivers, remote-sensing-based data (e.g., water surface elevation from SWOT and NDWI from RapidEye's images), and field measurements conducted with an Acoustic Doppler Current Profile (ADCP). The results show satisfactory performance of the model compared to the references. Based on the simulations, the authors conclude that the Jacuí and Taquiri rivers are the main contributors to the flooding in May 2024.

The manuscript shows promising results, but I have several comments that need to be addressed (e.g., methodological improvements, restructuring of some sections, etc.) before the manuscript can be considered for publication. My main concern is that the manuscript's goal is not fully addressed, and the experimental design does not allow conclusions to be drawn regarding enhancing our understanding of flooding mechanisms in South Brazil. I hope my comments and feedback help the authors highlight the great work they have done so far.

**Main comments:**

In the following points, I summarize the observations derived from my review of the paper, all of which aimed to improve the author's contribution.

- Take-home messages are missing in the abstract. For example, the authors mentioned "major lessons" but without providing further thoughts about them. I recommend including some of the main conclusions of the study.
- In the introduction, an improvement in the literature review is needed to highlight the relevance of this topic and the local context. Here are some points that I am missing: (i) an overview of what is known about flood generation mechanisms in Brazil, (ii) highlight the need to improve flood modeling and keep the consistency in the examples provided (e.g., you mentioned LISFLOOD with a UK flood and the other examples are focus on Brazil- which I considered reasonable and consistent with the study goals), (iii) frame the manuscript as a case study (which is clearer), and some thoughts about to what extent their results are potentially extrapolatable to other regions (this last point should be revisited later in the discussion).
- Continuing with the introduction, I recommend shortening some paragraphs to improve the clarity of the document. The paragraph related to climate change can easily be combined with paragraph one and shortened to 2-3 sentences, as the focus of this manuscript is not climate change, but rather the event of May 2025. In this context, climate change is a motivation (or rather a concern) to improve our understanding of extreme events in a changing climate. To further complement this necessity, you could also include a brief mention of the concepts being discussed today in the community related to compound events (e.g., Heinrich et al., 2023; Hendry et al., 2019, Leonard et al., 2014), hydroclimatic volatility (e.g., Swain et al., 2025), and hydrological volatility (e.g., Hammond et al., 2025).
- After reading the manuscript several times, I think that the objective of the document is misleading, particularly for the use of the word "mechanisms". Studies analyzing flooding mechanisms focus on, e.g., interactions of hydrometeorological processes, dominant processes, their relationship with flooding magnitude and timing, among others (see, e.g., the study by Jiang et al. (2022) and some of those cited in its introduction). Here, only the contribution of streamflow from different tributaries is being considered, without a deeper understanding of the processes occurring in each of them. In this context, questions arise such as: How sensitive is the response of each catchment to changes in precipitation/temperature,

and how does this propagate downstream (both in magnitude and timing)? When does the regulating effect of the catchment not play a key role anymore? Under what conditions can peak flow synchronization occur? Based on your results, it is clearer to me that the questions you are addressing are those that arose regarding the function of the natural system after the disaster (referring to Hunt et al., 2024; Silva et al., 2024a).

- Following on from the previous point, my recommendation is: (i) rewrite the objective of the study and align it with the questions that arose after the May 2025 flood, or (ii) deepen the analysis to improve understanding of the processes and interactions that shape the characteristics of the flood under study.

- How stable is the riverbed of the rivers studied? I suggest improving the discussion regarding the representativity of the bathymetry (i.e., changes in the riverbed).

- Regarding the databases used, is there information that allows for the uncertainty of each one to be incorporated? This is especially important for products based on remote sensing. However, for altimetry, it would also be important to have information related to the discharge curve associated with each station and its characteristics (i.e., maximum height) to understand the scope of the extrapolated records.

- In the flood modeling part, it is not clear to me how the authors are including the intermediate contribution of water to the flood. In other words, how are contributions to the modeling grid considered? I understand that the streamflow input and output are fixed boundary conditions, but are the rest of the values purely given by the numerical closure of the simulations?

- When calibrating the model, what is calibrated? Only Manning's roughness? How is this calibration performed? Is there an a priori spatial distribution assumption, and then a super-parameter is calibrated to reduce dimensionality (i.e., regularization)? How sensitive are the results to the selected parameters?

- The explanation of the experiments could be improved a little. It isn't easy to follow the experiments and then the scenarios (which contain each other). Additionally, what justifies these scenarios? How feasible are they? How is the downstream influence that the ocean could have on the channels designed to discharge there?

- Results associated with water level simulation are challenging to interpret. I don't know how different the damages can be if we have a bias of 1 or 10 m. I recommend that the authors refer to the level, maybe to, e.g., the riverbanks, to present the results in terms of river flooding potential (or include a line in the plots showing the riverbanks' height). This could help to highlight the results from a hazard perspective.

- In Figure 4, instead of presenting 12 panels, why not show two examples (panels a-b) and then a third panel (c) with a box plot showing the absolute error in time in each of the simulations? Or maybe a plot showing the NSE, RMSE, and BIAS. Note that the use of NSE provides the same information as RMSE. Additionally, how are differences in level translated into total flood volume and maximum flows?

- Considering the availability of a gridded product for water level (SWOT), why was it not considered to present a map of differences with the simulations?

- The flood extension figure (Figure 6) does not allow for analysis of the results. The base map makes it difficult to distinguish the blue lines representing the HEC-RAS simulation. Within each panel, it would be very informative to include the PlanetScope area, HEC-RAS (blue color), and the difference between the two.

- The verification of streamflow (Figure 7) should also be done with the streamflow series recorded by the stations shown in Figure 2. In addition, uncertainty bands should be included in ADCP measurements to make the comparison fairer.

- Figure 8 could be improved by changing the focus of the analysis. Instead of removing one tributary at a time, I think that testing each one independently (by "turning off" the rest) would provide more information. This is because the sum of the tributaries (routed to the control point) should be equivalent to the observed flood event. With that, you can have a stacked area chart where, for each time step, you know the relative contribution of each of the basins.

As a reference to what I meant (applied in a different context, not related to floods), you can see Figure 7 in Ayala et al. (2020).

- How (physically) feasible are the river flood synchronization scenarios? I think this scenario is exciting, but it would be good to explore the likelihood of this happening in more depth (I hypothesize that it would be closely related to the type of storm and its spatial distribution).
- As the results showed that the proposed hydraulic interventions would have a limited benefit, why don't the authors remove the analysis from the main manuscript? It is unclear how these scenarios are formulated or how feasible they are in technical, economic, and other terms (I suppose certain environmental agencies would raise concerns about the construction of a channel connecting the lagoon to the ocean). To better understand the proposed modifications, it would be beneficial to justify them and explore alternative solutions that offer a significant benefit in alleviating flooding in the area.
- The clarity and readability of the manuscript could be improved by splitting the results section from the discussion. Currently, the description of the results is overshadowed by the discussion.
- The findings presented support points (i) and (ii) of the conclusions, but not the second sentence of point (iii) (L428-429). The low contribution of the proposed hydraulic solutions may be linked to the design, the characteristics of the flood, and other factors. There is insufficient information to conclude that location is a determining factor. I recommend rewriting that idea to clarify the point you are trying to make.
- The paragraph between L430-434 should be included in the discussion (limitations) rather than in the conclusions.
- The statements between L440-442 go beyond what is presented in the manuscript. What could be highlighted – instead of mentioning the idea of "serve as a benchmark" - is the incorporation of different sources of information for the evaluation of the modeling. I recommend rewriting this paragraph to highlight the need for verification and constraining parameters in numerical models, based on the incorporation of complementary information to enhance realism and fidelity in simulations.

**Minor comments:**

- In Figure 2, consider including the points where the ADCP measurements are available.
- For all the figures, check the readability of the labels and (maybe) consider reducing their 'multidimensionality' to guide the readers straight to the point you want to make.
- L42: "In instance" → **For** instance.
- L205: "Finally, a set of hydraulic intervention**s** experiments was **organized**" → Finally, hydraulic intervention experiments were tested.
- L299: "… the peak water would lower **xx** meters to 4.75 meters,…" → typo + verb is missing
- L417: "Our findings **address the following scientific questions**:…"→ Our findings are summarized as follow:

**References**

Ayala, Á., Farías-Barahona, D., Huss, M., Pellicciotti, F., McPhee, J., & Farinotti, D. (2020). Glacier runoff variations since 1955 in the Maipo River basin, in the semiarid Andes of central Chile. The Cryosphere, 14(6), 2005-2027.

Hammond, J., Anderson, B., Simeone, C., Brunner, M., Muñoz-Castro, E., Archfield, S., ... & Armitage, R. (2025). Hydrological Whiplash: Highlighting the Need for Better Understanding and Quantification of Sub-Seasonal Hydrological Extreme Transitions. Hydrological Processes, 39(3), e70113.

Heinrich, P., Hagemann, S., Weisse, R., Schrum, C., Daewel, U., & Gaslikova, L. (2023). Compound flood events: analysing the joint occurrence of extreme river discharge events and storm surges in northern and central Europe. Natural hazards and earth system sciences, 23(5), 1967-1985.

Hendry, A., Haigh, I. D., Nicholls, R. J., Winter, H., Neal, R., Wahl, T., ... & Darby, S. E. (2019). Assessing the characteristics and drivers of compound flooding events around the UK coast. Hydrology and Earth System Sciences, 23(7), 3117-3139.

Jiang, S., Bevacqua, E., & Zscheischler, J. (2022). River flooding mechanisms and their changes in Europe revealed by explainable machine learning. Hydrology and Earth System Sciences, 26(24), 6339-6359.

Leonard, M., Westra, S., Phatak, A., Lambert, M., van den Hurk, B., McInnes, K., ... & Stafford-Smith, M. (2014). A compound event framework for understanding extreme impacts. Wiley Interdisciplinary Reviews: Climate Change, 5(1), 113-128.

Swain, D. L., Prein, A. F., Abatzoglou, J. T., Albano, C. M., Brunner, M., Diffenbaugh, N. S., ... & Touma, D. (2025). Hydroclimate volatility on a warming Earth. Nature Reviews Earth & Environment, 6(1), 35-50.

---

## Author Response (AR1)

**Response letter**

Leonardo Laipelt, Fernando Mainardi Fan, Rodrigo Cauduro de Dias de Paiva, Matheus Sampaio, Walter Collischonn and Anderson Ruhoff

Dear Editor,

We are resubmitting the revised version of our manuscript, "*Mechanisms and scenarios of the unprecedented flooding event in South Brazil 2024*" which presents a hydrodynamic assessment and evaluation of the May 2024 flood in Southern Brazil.

We sincerely thank the reviewers for their comprehensive and insightful comments. We believe their feedback has significantly strengthened the paper's clarity, structure, and scientific impact. The core changes we implemented involve: A complete restructuring and rewriting of the manuscript to explicitly define the research questions and novelty; revisions to the Methods section, including a new workflow diagram and detailed design experiments; the addition of a dedicated Discussion section that compares our model's performance and results to existing literature and addresses uncertainties; substantial improvements to data visualization and overall readability.

We are confident that these extensive revisions have resulted in a robust scientific contribution that now meets the standards for publication.

Best,

Leonardo Laipelt On behalf of the authors

**RC1**

This study uses a 2D hydrodynamic modelling framework to evaluate the hydraulic mechanisms driving the 2024 flooding event in southern Brazil. First, an evaluation of the modelling approach is conducted using different data sources. The authors then perform modelling experiments to: determine which rivers contributed most to flooding in RMPA; understand the consequences of potential synchronous flooding in the two main rivers; and determine whether flood control measures could have reduced river levels. It is an interesting topic, and I understand that the authors put a lot of effort into evaluating their approach using different data sources. However, in my opinion, the paper is not well written and fails to explain how the research is novel, what the research questions are, and how the results and framework compare with those of other studies in the field. Furthermore, the main goals of the study are unclear and the methodology lacks the overarching structure required to achieve them. I provide more detailed comments

below to demonstrate this point. The manuscript would need to be reshaped and rewritten to make a valuable contribution to HESS.

- We sincerely appreciate the reviewer's thoughtful assessment and recognition of the effort invested in this study. We respectfully emphasize our strong belief in the scientific relevance and potential impact of our work, which addresses a record-breaking flooding event of exceptional socio-environmental significance in southern South America. Following the reviewer's guidance, we have substantially revised the manuscript to clarify the novelty of our approach, explicitly state the research questions and main objectives, and strengthen the methodological structure. We have also expanded the discussion to compare our results and framework with other studies in the field, ensuring that the contribution of our modelling experiments is clearly demonstrated. We believe these improvements significantly enhance the clarity, rigor, and value of the paper, and we are confident that the revised version now meets the standards expected for publication.

Detailed comments:

The introduction lacks clear structure and research questions derived from an overview of existing research in the field. The main messages that the authors want to convey in each paragraph are difficult to follow. For example:

- A lot of emphasis is placed on the effect of climate change on flood extremes (e.g. L33–46), despite this not being a topic addressed by the study.

Response: Thank you for pointing this out. We restructured our introduction based on our suggestions, clarifying our research questions and highlighting our objections. (i) We removed the emphasis on the effect of climate change in the introduction, substituting by the benefits of using hydrodynamic models and exist research regarding flood scenarios over urban areas; (ii) added the motivations for studying flood synchronization scenarios and potential mitigation measures within the context of the study area; (iii) address the main research questions raised by our study.

- The authors emphasise that hydrodynamic models are used for such studies and present existing applications (L66–78). However, they do not highlight what is missing or how the present study differs from or builds on these approaches.

Response: Thank you for your comment. We have revised the Introduction to clearly articulate the gap and our contribution. Specifically, we now (i) frame the May 2024 event as an exceptional, record-scale flood in southern Brazil that exposed limitations of prior applications focused on single rivers, sparse validation, or simplified boundary controls: (ii) explain that our study was made possible by integrating a comprehensive dataset – detailed bathymetry, ADCP discharge and velocity transects, continuous water-level records, and satellite-derived inundation maps – together with the use of SWOT satellite altimetry for model validation in this context; and (iii) show how this robustly

constrained 2D model enables controlled experiments on synchronous peals and on realistic flood-control scenarios across the river-estuary-lagoon system, providing real world knowledge on this system functioning and enabling actionable insights for agencies and stakeholders in the most affected areas. These additions clarify what was missing in earlier works (limited validation and system-wide counterfactual testing) and how our framework builds on and extends existing approaches.

- The main objective of the study is presented as follows: "understanding of flooding mechanisms in South Brazil" (L79), yet little is said beforehand to explain why this is necessary and why it has not been done before. Some explanations that are not central to the introduction are provided at line 61: "After the disaster, many questions were raised regarding the function of the natural system: the relevance of the upstream rivers, the slopes generated by water inflows and even if extra outlets in the lagoon to the sea would not have avoided the flooding at upstream areas (Hunt et al., 2024; Silva et al., 2024a).".

  **Response:** Thank you for pointing this out. We rewrote the Introduction to (i) motivate the necessity of the study before stating the objectives, and (ii) clarify why this has not been addressed at system scale in prior work. Specifically, we now explain that the May-2024 flood was an exceptional, basin-wide event that triggered competing hypotheses among agencies and the public (relative river contributions, the role of peak synchrony, and whether additional lagoon–ocean outlets could have mitigated upstream flooding). Addressing these questions requires an integrated estuary–lagoon–river modelling and multi-sensor validation framework that has not previously been available (detailed bathymetry, ADCP, continuous water levels, satellite inundation, and SWOT altimetry combined). We also have rephrased the objectives to emphasize the primary aim— understanding how the natural system functions both under extreme flood conditions and when perturbed by plausible structural interventions—while framing the study as a generalizable case for other coupled river–lake–lagoon systems. The specific analyses (river contributions, peak synchrony, and mitigation scenarios) are now presented as means to achieve this overarching objective rather than ends in themselves.

  Given these changes, now one can read in the early paragraphs and in the objective of the work the following sentences:

  *"This paper develops the first detailed hydrodynamic assessment of the unprecedented flood that occurred in 2024 in south Brazil, which represents the worst disaster in Brazilian history. In addition to this novelty, it is the first study to utilize SWOT satellite altimetry data for model validation. Our primary goals are to investigate the main mechanisms governing this flood disaster and to assess hydraulic intervention scenarios for flood control in the region, which are currently under public debate. To achieve this, we address urgent and unresolved questions raised by the May-2024 flood regarding: (a) the relative influence of tributary inflows on RMPA water levels and inundation, (b) the consequences of potential peak synchrony between the main rivers, and (c) whether additional lagoon–ocean outlets or channel operations would have mitigated upstream flooding.*

*Prior studies did not jointly address these system-scale dynamics due to limited integrated datasets and validation across the river–estuary–lagoon continuum. Leveraging detailed bathymetry, ADCP transects, continuous gauges, satellite flood extent, and SWOT altimetry (Biancamaria et al., 2016; Fu et al., 2024), we develop and validate a 2D hydrodynamic model to quantify mechanisms and test counterfactual scenarios. This design yields decision-relevant evidence for stakeholders and government agencies seeking to enhance protection in the most affected areas, and ultimately allows comprehension of how this unique natural system works under extreme conditions. The insights from this study are therefore highly relevant for other complex, large-scale hydrodynamic coastal and deltaic regions..”*

- This study analyses potential mitigation measures and flooding synchronicity, but the introduction provides no background to explain why this is relevant, what has been done to assess this in other studies, or how their approach or analyses are novel in that regard.

  **Response:** Thank you for this important point. We have revised the Introduction to establish: (i) why peak synchrony and mitigation measures are decision-relevant in large, coupled river–estuary–lagoon systems; (ii) what is known from prior work, noting that relatively few studies evaluate system-scale synchrony effects and counterfactual, hydraulically consistent mitigation scenarios across the full continuum; and (iii) how our approach is novel, namely by combining a basin-to-lagoon 2D hydrodynamic model with multi-sensor validation (bathymetry, ADCP, gauges, satellite inundation, and SWOT altimetry) to run controlled experiments that isolate synchrony mechanisms and quantify the potential (and limits) of structural interventions. This framing clarifies both the relevance and the innovation of our analysis.

  We added in the introduction the following sentence:

  *"For instance, these models are particularly useful for studying complex interactions in medium-to-large basins (O'Loughlin et al., 2020; Paiva et al., 2013), where precipitation is expected to become more concentrated. In these coupled systems, the synchrony between the peak flows of major tributaries and the estuary–lagoon water level is a primary determinant of flood severity, directly informing the timing and feasibility of structural and operational measures (Guse et al., 2020). While previous studies have often focused on individual rivers or local interventions (Dutta et al., 2007; Patel et al., 2017; Timbadiya et al., 2015; Zarzuelo et al., 2015), few have examined synchrony and mitigation within an integrated, river–estuary–lagoon framework at regional scale. Moreover, simulating flood mitigation scenarios is essential for evaluating interventions, defining optimal locations for new structures, assessing the efficiency of existing ones (Abdella and Mekuanent, 2021; Ghanbarpour et al., 2013; Zhang et al., 2021), and identifying areas of high risk (Cai et al., 2019; Li et al., 2019; Masood and Takeuchi, 2012)."*

I suggest that the authors completely reshape and rewrite the introduction to focus on their main analyses and questions, providing a clearer justification for their study. This could be achieved by focusing on four main apsects: 1) model evaluation using different sources of data,

2) the hydraulic mechanisms/drivers of flooding, 3) flooding synchronicity, and 4) the evaluation of mitigation measures.

**Response:** We are very grateful to the reviewer for their suggestions to improve our introduction. We agree that the current version lacks a clear narrative and will completely rewrite it as suggested. The new introduction establishes the importance of robust model evaluation and discusses the knowledge gaps regarding the hydraulic drivers of flooding in our study area. Additionally, we rephase the introduction to highlight the relevance of studying flooding synchronicity in the context of our study and introduce more details related to the need for evaluating mitigation measures. We are confident that with these alterations, the introduction will be significantly improved, enhancing clarity and understanding for the reader.

The method section lacks clear structure, making it difficult to follow. It would have been useful to include a diagram presenting an overview of the different experiments to help readers understand the study. Furthermore, many methodological points are introduced in the Results section, making it difficult to link the different experiments to the study's objective. For instance, the synchronisation experiment is only partially explained in section 4.2.2 of the results. While this experiment may seem trivial to some readers, I believe it would benefit from more detailed explanations of the exact methods employed.

**Response:** We appreciate this constructive suggestion and have substantially re-organized the Methods for clarity and reproducibility. First, we now provide a one-page schematic/flowchart that summarizes the workflow and the four experiment families (baseline simulations, river-contribution attribution, peak-synchrony sensitivity, and mitigation scenarios), indicating inputs, boundary conditions, and key outputs for each. Second, we moved all methodological content that was previously embedded in the Results (e.g., configuration details, boundary manipulations, evaluation metrics) into the Methods, so that each experiment is introduced before results are presented. Third, we expanded the synchronization experiment description to specify: (i) how upstream hydrographs are phase-shifted (advances/delays applied at the Taquari and Jacuí boundaries over a predefined range and regular increments, preserving hydrograph shape and volume); (ii) how control vs. perturbed runs are paired; (iii) which boundary conditions remain fixed (e.g., ocean/lagoon stage series) to isolate synchrony effects; (iv) model warm-start/spin-up procedure; and (v) evaluation metrics (changes in peak water level, peak timing, inundated area/depth, and gauge-based skill). This restructuring explicitly links each experiment to the overarching objective (understanding natural-system functioning under extreme floods and under plausible structural modifications) and should make the paper easier to follow.

Manuscript changes (Methods):

- Section 3.1 – Model domain, mesh, and parameters: domain extent (river–estuary–lagoon continuum), grid resolution, roughness parameterization, warm-start. Forcings and boundary conditions: upstream inflow hydrographs, lagoon/ocean levels, data assimilation choices (if any).
- Section 3.2 – Observational datasets and validation metrics: bathymetry sources, ADCP transects, gauge water levels, satellite inundation, SWOT altimetry; RMSE, bias, timing error, inundation overlap.

- Section 3.3 – Experiment design overview (new Figure – workflow diagram): matrix of experiments and outputs.
- 3.3.1 Baseline simulations: configuration and validation period.
- 3.3.2 River-contribution attribution: protocol for selectively scaling/holding inflows to quantify marginal effects on levels/inundation.
- 3.3.3 Peak-synchrony sensitivity (expanded): phase-shift protocol for Taquari/Jacuí hydrographs (regular time increments; volume-conserving shifts), fixed external BCs, pairing of runs, metrics reported.
- 3.3.4 Mitigation scenarios: representation of structural interventions (geometry/roughness or boundary adjustments), performance indicators, and trade-off assessment.

- Figures 10 and 11 are difficult to understand. The quality of the panels on the right is poor, the lines are thin and close together, and there are many sub-panels with little space between them. It is therefore difficult to understand how the figures can support the analyses presented.
  **Response:** Thank you for your feedback regarding the readability of Figures 10 and 11. We will revise the layout of these figures to more clearly illustrate the different hydraulic intervention scenarios.

- There is no distinct discussion section, which highlights that the research questions are unclear and the study has not been compared to existing literature. In order to justify the recommendations presented in Section 4.5, the authors must discuss their results in more depth and demonstrate how they have addressed their research questions.
  **Response:** Thank you for highlighting this. We have revised the manuscript to separate the Results (Section 4) and the Discussion (Section 5), allowing for a clearer focus on our research questions. Specifically, the discussion section has been significantly expanded. We added a detailed comparison of our results with the existing literature, assessing our model's performance against other studies of flood scenarios in sensitive areas. We also investigate the viability and efficiency of structural interventions by reviewing literature on hydraulic measures for flood control, exploring their benefits and limitations in the context of the May 2024 flood.

  We now proposed the following sections of discussion:
    o Section 5.1 - Model performance compared previous studies
    o Section 5.2 - Uncertainties regarding the two-dimensional model
    o Section 5.3 - Recommendations for flooding managements and strategies in the region

  The flood synchronisation experiment could be very interesting if the author provided more motivation. Why was this experiment conducted? Maybe I missed the reason somewhere. Is it physically 'reasonable'? Were these rivers sometimes synchronised for flooding in the historical period? What motivated the different methodological choices?
  **Response:** We appreciate your suggestion and have reshaped the manuscript to highlight our motivations regarding the flood synchronization experiment. This experiment was conducted to represent a flood severity condition that is reasonable for the study area, as the main rivers that compose this basin are geographically wellseparated. This means that the rainfall can reach these rivers at different times and in different amounts, which could, in a combined scenario, reproduce a synchronized propagation of the water peak.

Another reason for reproducing this scenario is that the water peak in cities like Porto Alegre (the capital, which was most affected by the flood) was close to the maximum limit of its flood protection system. This means that in a more severe scenario with the same amount of rainfall as the May 2024 flood, the protection limits could be reached, as demonstrated in our flood synchronization experiment.

Nevertheless, we have added more details regarding the motivations of the flood synchronization in the Introduction section, as described in previously answered questions.

- The accuracy of the model needs to be put into perspective. For example, how does "an average BIAS of -0.47 meters between the water level peak in the stations" (L228) relate to flooding in the Guaíba River, which has "an average depth of 2 meters" (L95)? Does this mean that, in some cases, the model would not produce simulations exceeding a certain impact threshold?

  **Response:** Thank you for highlighting this point. The accuracy of the model represents an error of around -9% compared to maximum water peak observed in the Guaíba River and -23% over long-term water levels, which is relative in accordance with currently studies using hydrodynamic 2D model at moderate to large basins. Moreover, the difference between minimum flood level and historic floods in the basin are much higher than the average bias found. For example, historic floods such as May 2024 (around 5.2 meters) and May 1951 (around 4.75 meters) are relatively higher than the minimum flood level (3 meters in the gauge station reference), thus, the simulation would prevent both worsened floods that occurred in the Guaíba River.

  This is a relevant discussion for our study, and we have incorporated in the Section 5.1 regarding model performance.

- The authors mention that the Manning coefficient was calibrated: L148: "Initial values of Manning's roughness coefficient were derived from the literature, followed by manual calibration for the study period to ensure optimal accuracy.". The authors mention this aspect as a potential source of uncertainty in Section 4.4. Shouldn't a sensitivity analysis be performed outside the calibration period to evaluate the transferability of the results to other periods and flood events? Tuning the parameters could make the model more accurate for this flood event by compensating for other sources of uncertainty.

  **Response:** We did not perform a sensitivity analysis beyond the evaluated period due to difficulties with data availability for validation and the potential non-representativeness of the water level variation for this extreme event. Thus, we tuned the simulation parameters within the same validation period. While we agree that a sensitivity analysis of the Manning's coefficient would be relevant to the study, we did not include it in the main manuscript. We added it to a supplementary material for a clearer explanation of the selected parameters.

- I noticed many typos and issues with the way things were worded. I am not a native speaker but these issues sometimes made the text difficult to read. I have listed a few examples below:
  - "Flood becomes a major concern" L32
  - "However, the relationship between climate change and flood is complex, with impacts vary regionally and influenced by multiple factors" L36-37
  - "in instance" L42
  - "as consequence" L49
  - "wate" L101
  - "manually" L119
  - "The first main result is the model validation itself, which calculate values were compared to level gauges…" L201 needs to be reformulated.
  - "We accessed" L309
  - "testes" L311
  - "from studies as flood mitigation measures" L326
  - "The analysis was based using 2D hydrodynamic modelling" L415

**Response:** We thank the reviewer for this feedback and apologize for the errors. We will carefully proofread the entire manuscript to correct all spelling and grammatical mistakes.

**RC2**

The authors examine the May 2024 flood in the Patos Lagoon basin in southern Brazil by implementing HEC-RAS 2D. The hydrodynamic model is calibrated and evaluated by comparing their simulations with water level records from gauge stations along the tributary rivers, remote-sensing-based data (e.g., water surface elevation from SWOT and NDWI from RapidEye's images), and field measurements conducted with an Acoustic Doppler Current Profile (ADCP). The results show satisfactory performance of the model compared to the references. Based on the simulations, the authors conclude that the Jacuí and Taquiri rivers are the main contributors to the flooding in May 2024. The manuscript shows promising results, but I have several comments that need to be addressed (e.g., methodological improvements, restructuring of some sections, etc.) before the manuscript can be considered for publication. My main concern is that the manuscript's goal is not fully addressed, and the experimental design does not allow conclusions to be drawn regarding enhancing our understanding of flooding mechanisms in South Brazil. I hope my comments and feedback help the authors highlight the great work they have done so far. Main comments: In the following points, I summarize the observations derived from my review of the paper, all of which aimed to improve the author's contribution.

**Response:** We sincerely thank the reviewer for their constructive suggestions and effort, which have helped improve our study. We also appreciate their recognition of our work's scientific contribution.

Based on this valuable feedback, we have substantially revised the manuscript to enhance its clarity and better highlight our research questions and main objectives. Specifically, we restructured the Introduction to more clearly present our study's motivations and goals. We also expanded the Methodology section, adding new subsections to detail our analysis of

synchronization and the proposed intervention structures. Furthermore, we have separated the Results and Discussion into two distinct sections, significantly expanding the latter to compare our findings with existing literature and better demonstrate the impact of our work.

We are confident that these revisions have significantly strengthened the manuscript and that they now meet the high standards required for publication.

• Take-home messages are missing in the abstract. For example, the authors mentioned "major lessons" but without providing further thoughts about them. I recommend including some of the main conclusions of the study.

**Response:** Thank you for your suggestion. We have revised the abstract and included more takeaway messages to better contextualize the manuscript. The abstract now reads as follows:

*"In May 2024, an extraordinary precipitation event triggered record floods in southern Brazil, particularly impacting complex river-estuary-lagoon systems, and resulting in unprecedented impacts on the local population and infrastructure. As climate change projections indicate an increase in such events for the region, understanding these flooding processes is essential for better preparing cities for future events like the May 2024 flood. In this context, hydrodynamic modelling is an important tool for reproducing and analysing this past extreme event. This paper presents the first detailed hydrodynamic assessment of this unprecedented flood, the worst registered natural disaster in Brazilian history. We also performed the first validation of a detailed hydrodynamic model using new observations from the SWOT satellite. The study investigates the main mechanisms that governed the disaster and assesses scenarios for hydraulic flood control interventions currently under public debate, with a focus on the most populated areas of the Metropolitan region of Porto Alegre (RMPA) capital city. The results demonstrated that the model accurately represented the event, with average NSE, RMSE and BIAS of 0.82, 0.71 meters and -0.47 meters, respectively, across the basin's main rivers. Furthermore, the simulated flood extent showed an 83% agreement with high-resolution satellite images. Our analysis of the governing mechanisms showed that the Taquari River was mainly responsible for the peak in the RMPA, while the Jacuí River contributed most to the flood's duration. Additionally, the synchronization of the flood peaks from both rivers could have increased water levels by 0.82 meters. Evaluated hydraulic interventions demonstrated that the effectiveness of the proposed measures varied by location, with a generally limited influence on RMPA water levels (lower than 0.38 m). By accurately assessing the May 2024 flood, this study enhances the understanding of a complex river-estuary-lagoon system, quantifies the impacts of adverse scenarios, and reveals the limitations of potential hydraulic structure interventions. Finally, modelling this unprecedented event offers valuable insights for future research and global flood management policies."*

• In the introduction, an improvement in the literature review is needed to highlight the relevance of this topic and the local context. Here are some points that I am missing: (i) an overview of what is known about flood generation mechanisms in Brazil, (ii) highlight the need to improve flood modeling and keep the consistency in the examples provided (e.g., you mentioned LISFLOOD with a UK flood and the other examples are focus on Brazil- which I considered reasonable and consistent with the study goals), (iii) frame the manuscript as a case study (which is clearer), and some thoughts about to what extent their results are potentially extrapolatable to other regions (this last point should be revisited later in the discussion).

**Response:** We thank the reviewer for their relevant thoughts. We agree that the introduction needs to be improved, and the points raised are important and relevant to our study. We have revised the Introduction to improve the literature review and contextualization of our study, and we also structure our research paper as more a case study. Specifically, we now (i) revisit previous studies on flood generation mechanisms in Brazil; (ii) highlight the potential of hydrodynamic models for evaluating flooding scenarios and the efficiency of possible and actual flood protection structures; and (iii) indicating in the introduction that the insights from this study are therefore highly relevant for other regions that are complexly controlled by multiple systems, such as river-estuary-lagoons.

We updated the manuscript with the following paragraphs:

Overview of flood generation mechanisms in Brazil:

"[…] *Floods in southern Brazil, situated in the sub-tropical and temperate portions of South America, have increased significantly in recent decades, a trend that has been supported by both historical data and climate projections (Ávila et al., 2016; Bartiko et al., 2019; Brêda et al., 2023; Chagas et al., 2022). Nationally, flood generation in Brazil is driven by a variety of mechanisms. These include intense convective storms causing urban flash floods (Cavalcante et al., 2020; Lima and Barbosa, 2019; Marengo et al., 2023), persistent rainfall associated with South Atlantic Convergence Zone (SACZ) leading to large-scale riverine floods, and the influence of major teleconnections like the El Niño-Southern Oscillation (ENSO). Specifically, in the southern region, the primary drivers are often intense frontal systems that bring widespread and prolonged precipitation (Ávila et al., 2016; Damião Mendes and Cavalcanti, 2014). Moreover, climate change is intensifying this scenario by increasing hydroclimate and hydrological volatility and altering flood-generating mechanisms (Hammond et al., 2025; Stevenson et al., 2022; Swain et al., 2025). This, in turn, increases the frequency and severity of floods, particularly through compound events (Heinrich et al., 2023; Hendry et al., 2019; Leonard et al., 2014) […]".*

Regarding potential of hydrodynamic models:

"*[…] For instance, these models are particularly useful for studying complex interactions in medium-to-large basins (O'Loughlin et al., 2020; Paiva et al., 2013), where precipitation is expected to become more concentrated. In these coupled systems, the synchrony between the peak flows of major tributaries and the estuary–lagoon water level is a primary determinant of flood severity, directly informing the timing and feasibility of structural and operational measures (Guse et al., 2020). While previous studies have often focused on individual rivers or local interventions (Dutta et al., 2007; Patel et al., 2017; Timbadiya et al., 2015; Zarzuelo et al., 2015), few have examined synchrony and mitigation within an integrated, river–estuary–lagoon framework at regional scale. Moreover, simulating flood mitigation scenarios is essential for evaluating interventions, defining optimal locations for new structures, assessing the efficiency of existing ones (Abdella and Mekuanent, 2021; Ghanbarpour et al., 2013; Zhang et al., 2021), and identifying areas of high risk (Cai et al., 2019; Li et al., 2019; Masood and Takeuchi, 2012). […]*

Potential to extrapolate for other regions:

*[…] The insights from this study are therefore highly relevant for other complex, large-scale hydrodynamic coastal and deltaic regions. […]*

• Continuing with the introduction, I recommend shortening some paragraphs to improve the clarity of the document. The paragraph related to climate change can easily be combined with paragraph one and shortened to 2-3 sentences, as the focus of this manuscript is not climate change, but rather the event of May 2025. In this context, climate change is a motivation (or rather a concern) to improve our understanding of extreme events in a changing climate. To further complement this necessity, you could also include a brief mention of the concepts being discussed today in the community related to compound events (e.g., Heinrich et al., 2023; Hendry et al., 2019, Leonard et al., 2014), hydroclimatic volatility (e.g., Swain et al., 2025), and hydrological volatility (e.g., Hammond et al., 2025).

**Response:** We appreciate the reviewer's further suggestions for our manuscript. We agree with the points raised and have reshaped the second and third paragraphs of the manuscript by shortening them and highlighting key points. The revised paragraphs are as follows:

*"Floods in southern Brazil, situated in the sub-tropical and temperate portions of South America, have increased significantly in recent decades, a trend that has been supported by both historical data and climate projections (Ávila et al., 2016; Bartiko et al., 2019; Brêda et al., 2023; Chagas et al., 2022). Nationally, flood generation in Brazil is driven by a variety of mechanisms. These include intense convective storms causing urban flash floods (Cavalcante et al., 2020; Lima and Barbosa, 2019; Marengo et al., 2023), persistent rainfall associated with South Atlantic Convergence Zone (SACZ) leading to large-scale riverine floods, and the influence of major teleconnections like the El Niño-Southern Oscillation (ENSO). Specifically, in the southern region, the primary drivers are often intense frontal systems that bring widespread and prolonged precipitation (Ávila et al., 2016; Damião Mendes and Cavalcanti, 2014). Moreover, climate change is intensifying this scenario by increasing hydroclimate and hydrological volatility and altering flood-generating mechanisms (Hammond et al., 2025; Stevenson et al., 2022; Swain et al., 2025). This, in turn, increases the frequency and severity of floods, particularly through compound events (Heinrich et al., 2023; Hendry et al., 2019; Leonard et al., 2014)."*

• After reading the manuscript several times, I think that the objective of the document is misleading, particularly for the use of the word "mechanisms". Studies analyzing flooding mechanisms focus on, e.g., interactions of hydrometeorological processes, dominant processes, their relationship with flooding magnitude and timing, among others (see, e.g., the study by Jiang et al. (2022) and some of those cited in its introduction). Here, only the contribution of streamflow from different tributaries is being considered, without a deeper understanding of the processes occurring in each of them. In this context, questions arise such as: How sensitive is the response of each catchment to changes in precipitation/temperature, and how does this propagate downstream (both in magnitude and timing)? When does the regulating effect of the catchment not play a key role anymore? Under what conditions can peak flow synchronization occur? Based on your results, it is clearer to me that the questions you are addressing are those that arose regarding the function of the natural system after the disaster (referring to Hunt et al., 2024; Silva et al., 2024a).

**Response:** We sincerely thank the reviewer for the comment, and we agree that the objective of the document is not clear. We have revised the manuscript and inserted in the introduction the main objective of our study for clarification, as detailed in the previous revisions. Regarding the use of the word "mechanisms", we are here referred of the flood processes thar main control the basin and how they could be different from a hydrodynamic perspective, including the flood

peaks synchronicity and the relevance of rivers. We acknowledge that the word "mechanisms" was used before to refer to other process. We also understand that this previous usage is not excluded from the processes that we have assessed.

Especially regarding the question "Under what conditions can peak flow synchronization occur?" the answer is relatively simple. The occurrence of precipitations in one region of the watershed and at another could be delayed from a meteorological point of view. We investigated this delay in our study, but looking at the flow routing from rivers.

• Following on from the previous point, my recommendation is: (i) rewrite the objective of the study and align it with the questions that arose after the May 2025 flood, or (ii) deepen the analysis to improve understanding of the processes and interactions that shape the characteristics of the flood under study.

**Response:** We thank the reviewer for their contributions. We rewrite the study's objectives as follows:

*"This study develops the first detailed hydrodynamic assessment of the unprecedented flood that occurred in 2024 in south Brazil, which represents the worst disaster in Brazilian history. In addition to this novelty, it is the first study to utilize SWOT satellite altimetry data for model validation. Our primary goals are to investigate the main mechanisms governing this flood disaster and to assess hydraulic intervention scenarios for flood control in the region, which are currently under public debate. To achieve this, we address urgent and unresolved questions raised by the May-2024 flood regarding: (a) the relative influence of tributary inflows on RMPA water levels and inundation, (b) the consequences of potential peak synchrony between the main rivers, and (c) whether additional lagoon–ocean outlets or channel operations would have mitigated upstream flooding.*

*Prior studies did not jointly address these system-scale dynamics due to limited integrated datasets and validation across the river–estuary–lagoon continuum. Leveraging detailed bathymetry, ADCP transects, continuous gauges, satellite flood extent, and SWOT altimetry (Biancamaria et al., 2016; Fu et al., 2024), we develop and validate a 2D hydrodynamic model to quantify mechanisms and test counterfactual scenarios. This design yields decision-relevant evidence for stakeholders and government agencies seeking to enhance protection in the most affected areas, and ultimately allows comprehension of how this unique natural system works under extreme conditions. The insights from this study are therefore highly relevant for other complex, large-scale hydrodynamic coastal and deltaic regions."*

• How stable is the riverbed of the rivers studied? I suggest improving the discussion regarding the representativity of the bathymetry (i.e., changes in the riverbed).

**Response:** Changes to the riverbeds of the analyzed rivers appear to have a negligible impact on the study's results, as they have, on average, remained largely stable over the years. Specifically, a report (Collischonn et al., 2025) compared bathymetry before and after the flood and found only minor changes (on the order of 10 centimeters). These morphological changes are insignificant when compared to the magnitude of the flood, which involved water level rises exceeding 30 meters.

• Regarding the databases used, is there information that allows for the uncertainty of each one to be incorporated? This is especially important for products based on remote sensing. However, for altimetry, it would also be important to have information related to the discharge curve associated with each station and its characteristics (i.e., maximum height) to understand the scope of the extrapolated records.

**Response:** We do not have all the necessary information regarding the uncertainty of each data incorporated in the simulation, whereas the discharge curve for the weather stations does not necessarily have the necessary data/or are updated for newest data. Nevertheless, we will incorporate this relevant topic into the discussion, regarding the uncertainties of each data (SWOT, satellite-derived) and addressing the implication of discharge uncertainty in our model forcing and how this could potentially affect our results.

For instance, we include following the preliminary assessment of the rating curves from main river stations in the watershed. One can see that some of the flows are from the extrapolation stretches of the curve, as asked:

[Figure]

*The reference for these curves is a report from Paiva et al. (2025), and we incorporated in the discussion the following sentence regarding the uncertainties of the observation data "On the other hand, it is important to note that there are also uncertainties regarding the observation data used for validation. A report by Paiva et al. (2025) indicated uncertainties in the extrapolation of the rating curve, particularly in locations such as Rio Pardo on the Jacuí River and at points along the Taquari River, which are localities that contributed significantly to the flood formation. These uncertainties in the stage-discharge relationship for extreme flows may explain some of the localized differences observed between the simulated and observed water levels."*

• In the flood modeling part, it is not clear to me how the authors are including the intermediate contribution of water to the flood. In other words, how are contributions to the modeling grid considered? I understand that the streamflow input and output are fixed boundary conditions, but are the rest of the values purely given by the numerical closure of the simulations?

**Response:** The model uses fixed boundary conditions. Upstream, the contributions from the main rivers are defined using information from gauge stations. The downstream boundary condition, in contrast, uses tidal data, as the system connects to the ocean. We did not add details on precipitation for the minor tributaries, as their contributions would be negligible compared to the flows from the main rivers. We will, however, add more details about the simulation setup in the Methods section to improve clarity.

• When calibrating the model, what is calibrated? Only Manning's roughness? How is this calibration performed? Is there an a priori spatial distribution assumption, and then a superparameter is calibrated to reduce dimensionality (i.e., regularization)? How sensitive are the results to the selected parameters?

**Response:** When we refer to the calibration of the model, we are referring to the definition of the manning roughness. We tested different manning values of each river reach, based on what we found in the literature, trying to find the optimum combination of the manning of those main rivers compared to its performance. We believe that including more details related to the Mannings roughness calibration would be benefit for the study, and we would be able to show how sensitive are the results. Overall, the manning values showed medium to high sensitivity to final results.

• The explanation of the experiments could be improved a little. It isn't easy to follow the experiments and then the scenarios (which contain each other). Additionally, what justifies these scenarios? How feasible are they? How is the downstream influence that the ocean could have on the channels designed to discharge there?

**Response:** We agree that the presentation of our experimental design can be significantly improved for clarity and that the justification and potential limitations of the scenarios need to be more explicit. We have substantially revised the manuscript to address these points. First, we added an Experiment design overview (new Figure – workflow diagram) in the manuscript, to provide a clear overview of the experiments. Second, we have reorganized the method section as follows:

- Section 3.1 – Model domain, mesh, and parameters: domain extent (river–estuary–lagoon continuum), grid resolution, roughness parameterization, warm-start. Forcings and boundary conditions: upstream inflow hydrographs, lagoon/ocean levels, data assimilation choices (if any).
- Section 3.2 – Observational datasets and validation metrics: bathymetry sources, ADCP transects, gauge water levels, satellite inundation, SWOT altimetry; RMSE, bias, timing error, inundation overlap.
- Section 3.3 – Experiment design overview (new Figure – workflow diagram): matrix of experiments and outputs.
- 3.3.1 Baseline simulations: configuration and validation period.
- 3.3.2 River-contribution attribution: protocol for selectively scaling/holding inflows to quantify marginal effects on levels/inundation.
- 3.3.3 Peak-synchrony sensitivity (expanded): phase-shift protocol for Taquari/Jacuí hydrographs (regular time increments; volume-conserving shifts), fixed external BCs, pairing of runs, metrics reported.
- 3.3.4 Mitigation scenarios: representation of structural interventions (geometry/roughness or boundary adjustments), performance indicators, and trade-off assessment.

We also added more justification in the Introduction, explicating that "[…] *This study develops the first detailed hydrodynamic assessment of the unprecedented flood that occurred in 2024 in south Brazil, which represents the worst disaster in Brazilian history. In addition to this novelty, it is the first study to utilize SWOT satellite altimetry data for model validation. Our primary goals are to investigate the main mechanisms governing this flood disaster and to assess hydraulic intervention scenarios for flood control in the region, which are currently under public debate. […]".*

Finally, we incorporated a description of out model's boundary conditions in the Methods section: "*The downstream boundary condition used tidal level time series from the Brazilian coast monitoring system (SIMCosta) network (https://simcosta.furg.br/home, accessed in August, 2024), located at the Patos Lagoon mouth. This ensures that the representation of the water levels over the basin are realistically subjected to the backwater effects of the tides under variable marine conditions. […]*"

• Results associated with water level simulation are challenging to interpret. I don't know how different the damages can be if we have a bias of 1 or 10 m. I recommend that the authors refer to the level, maybe to, e.g., the riverbanks, to present the results in terms of river flooding potential (or include a line in the plots showing the riverbanks' height). This could help to highlight the results from a hazard perspective.

**Response:** Thank you for this comment. We agree completely that the practical significance of the water level results is not immediately clear without proper physical reference. Presenting the data showing riverbanks' reference is an excellent idea that will substantially improve the manuscript.

• In Figure 4, instead of presenting 12 panels, why not show two examples (panels a-b) and then a third panel (c) with a box plot showing the absolute error in time in each of the simulations? Or maybe a plotshowing the NSE, RMSE, and BIAS. Note that the use of NSE provides the same information as RMSE. Additionally, how are differences in level translated into total flood volume and maximum flows?

**Response:** Thank you for this constructive suggestion. We kept the panel with the 12 stations and the statistics metrics because we believe this information is relevant to the readers. Furthermore, modifications were made to improve the understanding of the presented result, such as the addition of the flood elevation at each station.

• Considering the availability of a gridded product for water level (SWOT), why was it not considered to present a map of differences with the simulations?

**Response:** Thank you for this suggestion. We agree that providing a map comparing the water surface elevation from SWOT and the hydrodynamic model would be beneficial for the manuscript's discussion. We added this information to the updated manuscript version.

• The flood extension figure (Figure 6) does not allow for analysis of the results. The base map makes it difficult to distinguish the blue lines representing the HEC-RAS simulation. Within each panel, it would be very informative to include the PlanetScope area, HEC-RAS (blue color), and the difference between the two.

**Response:** Thank you for your feedback. We updated Figure 6 to more clearly distinguish between the simulated flood extent and the extent observed via remote sensing data.

The verification of streamflow (Figure 7) should also be done with the streamflow series recorded by the stations shown in Figure 2. In addition, uncertainty bands should be included in ADCP measurements to make the comparison fairer.

**Response:** As presented in Paiva et al. (2025), there is an uncertainty in the station discharge values resulting from the extrapolation of the rating curve. We preferred to maintain the comparison exclusively against the discharge data observed by the ADCP, as these data are considered more accurate. However, we do not have the precise uncertainty of the equipment observations to include in the Figure.

Figure 8 could be improved by changing the focus of the analysis. Instead of removing one tributary at a time, I think that testing each one independently (by "turning off" the rest) would provide more information. This is because the sum of the tributaries (routed to the control point) should be equivalent to the observed flood event. With that, you can have a stacked area chart where, for each time step, you know the relative contribution of each of the basins. As a reference to what I meant (applied in a different context, not related to floods), you can see Figure 7 in Ayala et al. (2020).

**Response:** We thank the reviewer for this suggestion. However, the sum of the tributaries contributions does not represent the equivalent magnitude of this flood event observed

downstream. This difference occurs due to highly non-linear nature of the flood wave propagation process in the river network. They are governed by the velocity of the flood wave, which is dependent on channel geometry and roughness, meaning that individual peaks area attenuated as its dislocated to downstream. Furthermore, the temporary storage capacity of floodplains can influence the routing process.

• How (physically) feasible are the river flood synchronization scenarios? I think this scenario is exciting, but it would be good to explore the likelihood of this happening in more depth (I hypothesize that it would be closely related to the type of storm and its spatial distribution).

**Response:** This scenario is highly plausible because the two river basins are geographically distinct enough to experience different rainfall events. A large-scale storm system, or a particular sequence of storms, could cause their flood peaks to synchronize as they propagate downstream. In Southern Brazil, the occurrence of multiple cold fronts within a few days are nor rare. For instance, in the May 2024, two cold fronts of varying spatial extent and intensity passed over the state of Rio Grande do Sul between April 27 and May 2.

Now, we have added the sentence to clarify the reason of the analysis:

*"This analysis evaluated the combined flood impact of the Jacuí and Taquari rivers on the RMPA, considering the context of previous events. In May 2024, two cold fronts of varying spatial extent and intensity passed over the state of Rio Grande do Sul between April 27 and May 2. As a consequence, the synchrony between the peak flows of major tributaries and the estuary-lagoon water level is a primary determinant of flood severity, directly informing the timing and feasibility of structural and operational measures.*

*Given their distinct flow propagation times, we simulated a theoretical worst-case scenario by shifting the Jacuí River hydrograph (at Rio Pardo) forward by approximately 4 days to force their flood peaks arrive simultaneously. This synchronization allowed us to evaluate the potential consequences for the region's flood protection systems."*

• As the results showed that the proposed hydraulic interventions would have a limited benefit, why don't the authors remove the analysis from the main manuscript? It is unclear how these scenarios are formulated or how feasible they are in technical, economic, and other terms (I suppose certain environmental agencies would raise concerns about the construction of a channel connecting the lagoon to the ocean). To better understand the proposed modifications, it would be beneficial to justify them and explore alternative solutions that offer a significant benefit in alleviating flooding in the area. •

**Response:** Thank you for raising this point. We included this analysis because it has become an internationally highly relevant topic since the May 2024 flood in southern Brazil. There is an ongoing debate among the public and environmental agencies regarding the efficiency and feasibility of such measures. However, scientific studies on their applicability in our region are lacking. While some reports have pointed to them as a solution (Hunt et al., 2024; Silva et al., 2024) they do not include an in-depth analysis. International consultancy studies suggested, but not tested, these kind of solutions after local assessments (Lamoree et al., 2024), resting unclear for the decision makers and the broad scientific community (specially beyond hydrologists) the relevance of these kinds of measures.

We believe scientific production should be free of confirmation bias and not restricted to reporting only "successful" outcomes. It must also document when widely suggested ideas, whether proposed by specialists or non-specialists, do not achieve the intended effects. Presenting negative or null results is essential to stress-test assumptions, refine hypotheses, and prevent costly missteps in policy and engineering. By transparently evaluating a range of plausible options and clearly communicating both what works and what does not, research better supports decision-makers with evidence-based guidance, a deeper understanding of system behavior, and a more realistic appraisal of uncertainties and trade-offs.

Therefore, our study aims to provide a scientific benchmark for local agencies and governments. We identify the key hydrodynamic processes and evaluate potential solutions for improving the flood protection system.

We added a justification regarding the hydraulic interventions in the Method section:

"*We tested proposed mitigation interventions currently under public and environmental agencies debate. Specifically, these proposals, which have not yet been formally evaluated, suggest the construction of new channels to reduce regional water levels in RMPA (DRRS, 2024; Hunt et al., 2024). Although these projects are still in the conceptual stage, we used the 2D hydrodynamic model to test their potential effects. This exercise aims to better comprehend the dominant forces controlling the system's dynamics.*"

The clarity and readability of the manuscript could be improved by splitting the results section from the discussion. Currently, the description of the results is overshadowed by the discussion.

> **Response:** We totally agree with the reviewer, and we have revised the manuscript to separate the Results (Section 4) and the Discussion (Section 5), allowing for a clearer focus on our research questions.
> Also, the discussion section has been significantly expanded with more detailed comparison of our results with the existing literature and investigations regarding the viability and efficiency of structural interventions.
>
> We now proposed the following sections of results:
> - Section 4.1 - Model validation
>   - Water level
>   - Flood extent
>   - Streamflow
> - Section 4.2 – Hydraulic mechanism of the flood
>   - River flood contribution
>   - River flood synchronization
> - Section 4.3 Hydraulic interventions for flood control
>   - Jacuí Guaíba channel
>   - Patos Lagoon channel
>
> And for discussion:
> - Section 5.1 - Model performance compared previous studies
> - Section 5.2 - Uncertainties regarding the two-dimensional model
> - Section 5.3 - Recommendations for flooding managements and strategies in the region

• The findings presented support points (i) and (ii) of the conclusions, but not the second sentence of point (iii) (L428-429). The low contribution of the proposed hydraulic solutions may be linked to the design, the characteristics of the flood, and other factors. There is insufficient information to conclude that location is a determining factor. I recommend rewriting that idea to clarify the point you are trying to make.

**Response:** Thank you for this suggestion. We revised the support points and reshaped (iii) by follows:

*"(iii) The proposed hydraulic structures of additional channels alternatives would not have been sufficient to prevent RMPA flooding entirely. Our results also indicated that the degree of flood mitigation structures would not have been uniform across the RMPA. This spatial disparity in performance suggests that the limited overall impact may be linked to a combination of factors, including the specific design of the interventions, local hydrogeomorphic features, and the unprecedented magnitude of the flood event itself"*

• The paragraph between L430-434 should be included in the discussion (limitations) rather than in the conclusions. •

**Response:** Thank you for noting this. We've reformulated the discussion and the conclusion of the manuscript and removed that part of the text.

The statements between L440-442 go beyond what is presented in the manuscript. What could be highlighted – instead of mentioning the idea of "serve as a benchmark" - is the incorporation of different sources of information for the evaluation of the modeling. I recommend rewriting this paragraph to highlight the need for verification and constraining parameters in numerical models, based on the incorporation of complementary information to enhance realism and fidelity in simulations.

**Response**: Thank you for your suggestion. The final sentence has been rewritten as follows:

*"Finally, this research advances a methodological framework predicated on multi-source data integration for the robust performance assessment of hydrodynamic simulations. By incorporating multiple, independent observational datasets, we significantly enhanced the model's predictive accuracy and its fidelity in reproducing this flood event. We expect that the presented methods will serve as a reference for studies in other locations, as well as for analyses of the efficiency of structural measures for flood control."*

**Minor comments:**

• In Figure 2, consider including the points where the ADCP measurements are available.

**Response:** Thank you for your suggestion, we incorporated the ADCP measurement's location in Figure 2.

• For all the figures, check the readability of the labels and (maybe) consider reducing their 'multidimensionality' to guide the readers straight to the point you want to make.

**Response:** We will revise all the figures in the manuscript to improve their readability.

• L42: "In instance" → For instance.

**Response:** Thanks, we corrected this.

• L205: "Finally, a set of hydraulic interventions experiments was organized" → Finally, hydraulic intervention experiments were tested.

**Response:** Thank you for noting, we adjusted this in the manuscript.

• L299: "… the peak water would lower xx meters to 4.75 meters,…" → typo + verb is missing

**Response:** Thank you for noting, we adjusted this.

• L417: "Our findings address the following scientific questions:…"→ Our findings are summarized as follow:

**Response:** Thank you. We corrected this.

**References**

Ávila, A., Justino, F., Wilson, A., Bromwich, D., & Amorim, M. (2016). Recent precipitation trends, flash floods and landslides in southern Brazil. *Environmental Research Letters*, *11*(11), 114029. https://doi.org/10.1088/1748-9326/11/11/114029

Bartiko, D., Oliveira, D. Y., Bonumá, N. B., & Chaffe, P. L. B. (2019). Spatial and seasonal patterns of flood change across Brazil. *Hydrological Sciences Journal*, *64*(9), 1071–1079. https://doi.org/10.1080/02626667.2019.1619081

Biancamaria, S., Lettenmaier, D. P., & Pavelsky, T. M. (2016). The SWOT Mission and Its Capabilities for Land Hydrology. *Surveys in Geophysics*, *37*(2), 307–337. https://doi.org/10.1007/s10712-015-9346-y

Brêda, J. P. L. F., Cauduro Dias de Paiva, R., Siqueira, V. A., & Collischonn, W. (2023). Assessing climate change impact on flood discharge in South America and the influence of its main drivers. *Journal of Hydrology*, *619*, 129284. https://doi.org/https://doi.org/10.1016/j.jhydrol.2023.129284

Cavalcante, M. R. G., da Cunha Luz Barcellos, P., & Cataldi, M. (2020). Flash flood in the mountainous region of Rio de Janeiro state (Brazil) in 2011: part I—calibration watershed through hydrological SMAP model. *Natural Hazards*, *102*(3), 1117–1134. https://doi.org/10.1007/s11069-020-03948-3

Chagas, V. B. P., Chaffe, P. L. B., & Blöschl, G. (2022). Climate and land management accelerate the Brazilian water cycle. *Nature Communications*, *13*(1), 5136. https://doi.org/10.1038/s41467-022-32580-x

Collischonn, W., Fan, F. M., PAIVA, R. C. D. de, Sampaio, M., Buffon, F., & Moraes, S. R. (2025). Modificações do leito do rio Taquari e seu impacto sobre as inundações. *Porto Alegre: IPH-UFRGS*.

Damião Mendes, M. C., & Cavalcanti, I. F. A. (2014). The relationship between the Antarctic oscillation and blocking events over the South Pacific and Atlantic Oceans. *International Journal of Climatology*, *34*(3), 529–544. https://doi.org/https://doi.org/10.1002/joc.3729

Fu, L.-L., Pavelsky, T., Cretaux, J.-F., Morrow, R., Farrar, J. T., Vaze, P., Sengenes, P., Vinogradova-Shiffer, N., Sylvestre-Baron, A., Picot, N., & Dibarboure, G. (2024). The Surface Water and Ocean Topography Mission: A Breakthrough in Radar Remote Sensing of the Ocean and Land Surface Water. *Geophysical Research Letters*, *51*(4), e2023GL107652. https://doi.org/https://doi.org/10.1029/2023GL107652

Hunt, J. D., Silva, C. V., Fonseca, E., de Freitas, M. A. V., Brandão, R., & Wada, Y. (2024). Role of pumped hydro storage plants for flood control. *Journal of Energy Storage*, *104*, 114496. https://doi.org/https://doi.org/10.1016/j.est.2024.114496

Lamoree, B., Verwey, A., Glerum, P., & Bacellar, D. (2024, August). Dutch Disaster Risk Reduction & Surge Support (DRRS) Programme: Final report—Porto Alegre, Brazil. Netherlands Enterprise Agency (RVO). https://english.rvo.nl/files/file/2024-08/DRRS%20Porto%20Alegre%20-%20final%20report%2025%20August%202024%20EN_0.pdf

Lima, R. C. de A., & Barbosa, A. V. B. (2019). Natural disasters, economic growth and spatial spillovers: Evidence from a flash flood in Brazil. *Papers in Regional Science*, *98*(2), 905–925. https://doi.org/https://doi.org/10.1111/pirs.12380

Marengo, J. A., Alcantara, E., Cunha, A. P., Seluchi, M., Nobre, C. A., Dolif, G., Goncalves, D., Assis Dias, M., Cuartas, L. A., Bender, F., Ramos, A. M., Mantovani, J. R., Alvalá, R. C., & Moraes, O. L. (2023). Flash floods and landslides in the city of Recife, Northeast Brazil after heavy rain on May 25–28, 2022: Causes, impacts, and disaster preparedness. *Weather and Climate Extremes*, *39*, 100545. https://doi.org/https://doi.org/10.1016/j.wace.2022.100545

Rodrigo Paiva ; FAN, F.M. ; COLLISCHONN, WALTER ; MEDEIROS, M. S. ; OLIVEIRA, R. C. ; LIMA, S. G. ; CAMARGO, P. L. B. . Caracterização e modelagem das cheias de 2023 e 2024 no Rio Grande do Sul em escala regional. 2025.

Silva, R. A. G., Reis, R. C. S., Ramos, D. M., Belém, A. L., Puhl, E., & Manica, R. (2024). Análise de Abertura de Novo Canal de Maré na Lagoa dos Patos para Atenuação de Cheias no Rio Guaíba, RS. *II FLUHIDROS - Simpósio Nacional de Mecânica Dos Fluidos e Hidráulica e XVI ENES - Encontro Nacional de Engenharia de Sedimentos*.

---

## Author Response (AR2)

Response Letter Round2

Leonardo Laipelt, Fernando Mainardi Fan, Rodrigo Cauduro de Dias de Paiva, Matheus Sampaio, Walter Collischonn and Anderson Ruhoff

Dear Editor,

We are resubmitting the revised version of our manuscript, "Mechanisms and scenarios of the unprecedented flooding event in South Brazil 2024," following the minor suggestions provided by the reviewers.

We have addressed each of the comments in a point-by-point format below. We believe these final adjustments have further strengthened the clarity of the paper.

Thank you for your continued consideration of our work.

Best,

Leonardo Laipelt, On behalf of the authors.

**Report #1**

Dear authors,

Thank you for answering and addressing my comments. The motivation for this study, the explanations of the methodology and the different experiments, as well as the presentation of the results have substantially improved. I think that the manuscript now meets the requirements for publication in HESS. I have a few remaining minor/technical comments:

- Dear Reviewer, we appreciate your contributions toward enhancing the quality of this manuscript. We have revised the text in accordance with your minor comments and updated the document accordingly. Thank you.

- L91: "ADCP" -> explain the acronym at the first mention

- The complete nomenclature for the acronym has been included.

- Figs 1 and 3: increase plot size

- The plot size for the Figures 1 and 3 was increased.

- Section 3.3.3: I think that the explanations about whether it is physically possible to have the two tributaries synchronized are still missing. The only reference to this is "In May 2024, two cold fronts of varying spatial extent and intensity passed over the region between April 27 and May 2. As a consequence,…" (L223-225). Here, "as a consequence" is really not clear. Again, I think this is a very interesting and relevant analysis but some information about which meteorological conditions would lead to such an event is not clear from the text. If both tributaries are triggered

by the same atmospheric event, then synchronicity will depend on the routing/propagation times, which, if I understand well, are different between the two rivers.

- Thank you for your feedback. We agree that the physical feasibility of this scenario needed further explanation. We have refined the manuscript to justify the synchronization experiment more clearly, as follows:

  *[…] This analysis evaluated the combined flood impact of the Jacuí and Taquari rivers on the RMPA. Although these tributaries have distinct flow propagation times, their peaks can synchronize depending on the spatio-temporal distribution of rainfall. In May 2024, the region was impacted by a sequence of two cold fronts between April 27 and May 2. Such sequential atmospheric events can lead to peak synchronization if the first system triggers discharge in the slower-responding basin (Jacuí), while a subsequent system impacts the faster basin (Taquari) with a delay that matches the difference in their routing times.*

  *To evaluate the potential consequences of such a meteorological alignment for the region's flood protection systems, we simulated a theoretical worst-case scenario by manually advancing the upstream hydrograph, used as the boundary condition for Jacuí River (at Rio Pardo), by approximately 4 days to force their flood peaks arrive simultaneously. This synchronization allowed us to evaluate the potential consequences for the region's flood protection systems. […].*

- L236: where does the Manning's roughness value come from?

  - The Manning's roughness coefficient was derived from established literature for earthen channels. The uncertainties associated with this choice are addressed in the Discussion, where we demonstrate that this channel parameter has minimal impact on the overall experimental findings concerning flood mitigation.

- 4.1.1 and Fig 5: comment in the text the difference between SWOT and the observed and simulated lines for Corsan.

  - We added the following sentence:

    *"[…] The Corsan station proved to be an exception, as SWOT observations did not effectively capture water level during the event. […]"*

- L256: "BIAS", no capital letters needed.

  - We updated to "bias".

- Figures 11 and 12: the colour legend is missing.

  - Thank you for noted this. We have updated the figures with color legends.

- 4.3.2: Given the position of the channel (downstream of points C, D and E), is it theoretically possible to obtain a reduction in the rising limb and the peak for C and D, especially if nothing is simulated before the event (i.e. antecedent lagoon levels)? If no, then it shouldn't be expected

in the hypotheses for this experiment. Additionally, I am still wondering whether the small observed reductions could still be useful to reduce flooding impacts, not related to the peak but to the recession that might be a bit shorter.

- Thank you for your question. We used antecedent lagoon levels prior to the event for a simulation warm-up period to ensure stable initial conditions. In Section 4.3.2, we explain that the only effect of the structural intervention would be a slightly shorter recession period (approximately two days). However, it would not protect the main cities of the RMPA from an extreme event of the magnitude observed.

- L448: "13%": this number was not reported in the results section. Additionally, could a reduction of 13% still reduce the flooding impacts to a certain extent?

- The 13% figure is simply another way of representing the information presented in the results, specifically comparing baseline water levels with those observed after hydraulic interventions. This reduction is not significant in the context of such an extreme event, which remains the primary focus of this study.

- L455-457: do you have a reference for this statement (especially for the second part of the sentence)? If not, I would remove.

- We have incorporated in the statement the following references about Porto Alegre urban development and affect areas due to the flood:
  - Miranda, A. Floods and extension plans: discourse and projects in Southern Brazil. In International Planning History Society Proceedings (Vol. 2). Delft, The Netherlands: IPHS, 2016.Collischonn

  - Collischonn, W., Fan, F. M., Possantti, I., Dornelles, F., Paiva, R., Sampaio, M., Michel, G., Magalhães Filho, F. J. C., Moraes, S. R., Marcuzzo, F. F. N., Michel, R. D. L. M., Beskow, T. L. C., Beskow, S., Fernandes, E., Laipelt, L., Ruhoff, A., Kobiyana, M., Collares, L. G., Buffon, F., Duarte, E., Lima, S., Meirelles, F. S. C., and Allasia, D.: The exceptional hydrological disaster of April-May 2024 in southern Brazil, Revista Brasileira de Recursos Hídricos, 1, https://doi.org/10.1590/2318-0331.302520240119, 2025.

**Report #2**

I want to thank the authors for taking into account my comments on the first draft of the manuscript. The updated version of the manuscript has improved considerably in terms of clarity, presentation of results, and analysis.

Here are some additional comments that I hope will enable you to further improve the quality of your work:

- Dear Reviewer, we appreciate your insightful feedback, which has significantly enhanced the clarity and impact of our work. We have carefully incorporated your suggestions into the revised manuscript, as detailed in our point-by-point responses below.

- In Section 3, I recommend starting with the current content in 3.3. This provides an overview of the content of the other sections, so starting with this text will give the readers a clearer idea of how the methodological pieces fit together.

- Thank you for your suggestion. We have moved the content in section 3.3 to the beginning of the Material and methods (Section 3). The methodology overview is now located in Section 3.1, titled "Workflow overview".

- I still believe that Figure 9 (in the updated manuscript) could be more informative if the focus of the analysis were changed. Instead of removing one tributary at a time, test each independently by deactivating the others. In my initial suggestion, I highlighted routing streamflows to the outlet point, as I understand the catchment's regulatory effect during floods and the nonlinear processes that can occur. Since you are using a model, there is no problem with doing this, as it follows the same approach of turning tributaries on and off. This method would provide more clarity regarding the relative contribution of each tributary.

- Thank you for this suggestion. We understand that testing each tributary independently would isolate their individual contributions. However, we believe that the 'leave-one-out' approach a more physically meaningful message for flood management.
Because the system is highly nonlinear, the impact of a single tributary depends on the state of the lagoon as dictated by the other inflows. Removing one tributary simulates a mitigation scenario of how much the flood levels would actually drop if that specific river's contribution were attenuated. This approach highlights the marginal impact of each river within the context of a saturated system, which is more relevant for decision-makers than evaluating a river in isolation.

- Following your argument, it is not clear to me how the peaks are synchronised and justified. Are the peaks simply 'pulled' so that they coincide, or are adjustments made to the model so that they coincide with the characteristics of the event and the system? Or is a less favourable design condition being represented? I recommend improving the description of that case study a little.

- We have updated the explanation of the peak synchronization scenario to clarify the adjustments made to the model. The revised text is as follows:

*[…] This analysis evaluated the combined flood impact of the Jacuí and Taquari rivers on the RMPA. Although these tributaries have distinct flow propagation times, their peaks can synchronize depending on the spatio-temporal distribution of rainfall. In May 2024, the region was impacted by a sequence of two cold fronts between April 27 and May 2. Such sequential atmospheric events can lead to peak synchronization if the first system triggers discharge in the slower-responding basin (Jacuí), while a subsequent system impacts the faster basin (Taquari) with a delay that matches the difference in their routing times.*

*To evaluate the potential consequences of such a meteorological alignment for the region's flood protection systems, we simulated a theoretical worst-case scenario by manually advancing the upstream hydrograph, used as the boundary condition for Jacuí River (at Rio Pardo), by approximately 4 days to force their flood peaks arrive simultaneously. This synchronization allowed us to evaluate the potential consequences for the region's flood protection systems […].*

- Regarding the proposed interventions to alleviate floods, I agree that, as scientists, we must demonstrate what is and isn't effective. Furthermore, I am convinced that it is often the things that do not turn out as we expect that allow us to learn the most. However, setting aside philosophy, my comment was intended to justify these cases. Without solid justification, we could try any configuration that we know will fail from the outset, and, in that sense, the trivial solution does not contribute any real value or knowledge. I understand that these are some of the discussions currently taking place in Brazil, and the description in the text has improved considerably. Nevertheless, I strongly recommend including a few additional lines to justify these cases from a technical standpoint and to present the authorities' hypotheses. This would further highlight the importance of the findings you present in the manuscript for decision-makers.

- Thank you for your suggestions and for raising this discussion. We agree that it is essential to provide solid technical justifications for the analyses developed in the manuscript. Our scenarios are based on the discourse that emerged within the public and governmental communities following the event, alongside ongoing studies regarding technical implementations. To clarify our justification, we have updated the Introduction as follows:
  - *[…] After the disaster, significant public and technical debated emerged regarding the hydraulic drivers of the flood. Question focused on the relative influence of upstream rivers, the slopes generated by water inflows, and the restrictive nature of the lagoon's single outlet to the ocean. Specifically, public and governmental debates have hypothesized that additional artificial outlets could have mitigated flooding in upstream areas (Hunt et al., 2024; Silva et al., 2024a) […].*

  We believe that with this revision, readers and decision-makers will be better able to understand the background behind our research.

- It would be interesting to see how the flood areas change in Figures 9 and 10 for the cases considered. The flooding component is one of the added values of your study, and I think it could also be highlighted here to highlight the hydrological risk explored.

Thank you for your comment. We agree that incorporating flood area results would be valuable for the study. However, as the peak water levels did not change significantly,

the total inundated areas remained virtually unchanged across these scenarios. For this reason, we decided not to include the maps in the manuscript. Nonetheless, we have added a textual statement in the results section to highlight this finding

We added the following sentence in section 4.3.1:
*[…] would result in only minor reduction in water levels, with minimal impact on flood-prone areas. […]~*
*and Section 4.3.2:*
*[…] while having minimal impact on the maximum flow peak and the extent of flooding upstream in the Guaíba River. […]*

Minor suggestions:

- L233: Instead of "exercise", prefer experiments or assessment. Exercise sounds like something synthetic, and potential solutions are being evaluated here.

- Thank you. We have changed to 'assessment' in the sentence.

- L236: Indicate the type of channel with which the selected roughness can be associated (e.g., n = 0.02-> earth channel). Since hydrology and hydraulics are combined here, these small details must make sense to all readers.

- Thank you, we have updated the sentence as follows: "[…] *The channels were assigned a Manning's roughness of 0.02, which corresponds to standard values for earthen channels […]*"

- L259: "river's man channel in the DTM" -> river's main channel in the DTM

- Thank you for pointing this out. The typo has been corrected in the manuscript.